# Decentralized Deterministic Multi-Agent Reinforcement Learning

## Abstract

[Zhang, ICML 2018] provided the first decentralized actor-critic algorithm for multi-agent reinforcement learning (MARL) that offers convergence guarantees. In that work, policies are stochastic and are defined on finite action spaces. We extend those results to offer a provably-convergent decentralized actor-critic algorithm for learning deterministic policies on continuous action spaces. Deterministic policies are important in real-world settings. To handle the lack of exploration inherent in deterministic policies, we consider both off-policy and on-policy settings. We provide the expression of a local deterministic policy gradient, decentralized deterministic actor-critic algorithms and convergence guarantees for linearly-approximated value functions. This work will help enable decentralized MARL in high-dimensional action spaces and pave the way for more widespread use of MARL.

## 1  Introduction

Cooperative multi-agent reinforcement learning (MARL) has seen considerably less use than its single-agent analog, in part because often no central agent exists to coordinate the cooperative agents. As a result, decentralized architectures have been advocated for MARL. Recently, decentralized architectures have been shown to admit convergence guarantees comparable to their centralized counterparts under mild network-specific assumptions (see Zhang et al. [2018], Suttle et al. [2019]). In this work, we develop a decentralized actor-critic algorithm with deterministic policies for multi-agent reinforcement learning. Specifically, we extend results for actor-critic with stochastic policies (Bhatnagar et al. [2009], Degris et al. [2012], Maei [2018], Suttle et al. [2019]) to handle deterministic policies. Indeed, theoretical and empirical work has shown that deterministic algorithms outperform their stochastic counterparts in high-dimensional continuous action settings (Silver et al. [January 2014b], Lillicrap et al. [2015], Fujimoto et al. [2018]). Deterministic policies further avoid estimating the complex integral over the action space. Empirically this allows for lower variance of the critic estimates and faster convergence. On the other hand, deterministic policy gradient methods suffer from reduced exploration. For this reason, we provide both off-policy and on-policy versions of our results, the off-policy version allowing for significant improvements in exploration. The contributions of this paper are three-fold: (1) we derive the expression of the gradient in terms of the long-term average reward, which is needed in the undiscounted multi-agent setting with deterministic policies; (2) we show that the deterministic policy gradient is the limiting case, as policy variance tends to zero, of the stochastic policy gradient; and (3) we provide a decentralized deterministic multi-agent actor critic algorithm and prove its convergence under linear function approximation.

Submitted to 34th Conference on Neural Information Processing Systems (NeurIPS 2020). Do not distribute.

## 2 Background

Consider a system of $N$ agents denoted by $\mathcal{N} = [N]$ in a decentralized setting. Agents determine their decisions independently based on observations of their own rewards. Agents may however communicate via a possibly time-varying communication network, characterized by an undirected graph $\mathcal{G}_t = (\mathcal{N}, \mathcal{E}_t)$, where $\mathcal{E}_t$ is the set of communication links connecting the agents at time $t \in \mathbb{N}$. The networked multi-agent MDP is thus characterized by a tuple $(\mathcal{S}, \{\mathcal{A}^i\}_{i \in \mathcal{N}}, P, \{R^i\}_{i \in \mathcal{N}}, \{\mathcal{G}_t\}_{t \geq 0})$ where $\mathcal{S}$ is a finite global state space shared by all agents in $\mathcal{N}$, $\mathcal{A}^i$ is the action space of agent $i$, and $\{\mathcal{G}_t\}_{t \geq 0}$ is a time-varying communication network. In addition, let $\mathcal{A} = \prod_{i \in \mathcal{N}} \mathcal{A}^i$ denote the joint action space of all agents. Then, $P : \mathcal{S} \times \mathcal{A} \times \mathcal{S} \to [0, 1]$ is the state transition probability of the MDP, and $R^i : \mathcal{S} \times \mathcal{A} \to \mathbb{R}$ is the local reward function of agent $i$. States and actions are assumed globally observable whereas rewards are only locally observable. At time $t$, each agent $i$ chooses its action $a_t^i \in \mathcal{A}^i$ given state $s_t \in \mathcal{S}$, according to a local parameterized policy $\pi_{\theta^i}^i : \mathcal{S} \times \mathcal{A}^i \to [0, 1]$, where $\pi_{\theta^i}^i(s, a^i)$ is the probability of agent $i$ choosing action $a^i$ at state $s$, and $\theta^i \in \Theta^i \subseteq \mathbb{R}^{m_i}$ is the policy parameter. We pack the parameters together as $\theta = [(\theta^1)^\top, \cdots, (\theta^N)^\top]^\top \in \Theta$ where $\Theta = \prod_{i \in \mathcal{N}} \Theta^i$. We denote the joint policy by $\pi_\theta : \mathcal{S} \times \mathcal{A} \to [0, 1]$ where $\pi_\theta(s, a) = \prod_{i \in \mathcal{N}} \pi_{\theta^i}^i(s, a^i)$. Note that decisions are decentralized in that rewards are observed locally, policies are evaluated locally, and actions are executed locally. We assume that for any $i \in \mathcal{N}$, $s \in \mathcal{S}$, $a^i \in \mathcal{A}^i$, the policy function $\pi_{\theta^i}^i(s, a^i) > 0$ for any $\theta^i \in \Theta^i$ and that $\pi_{\theta^i}^i(s, a^i)$ is continuously differentiable with respect to the parameters $\theta^i$ over $\Theta^i$. In addition, for any $\theta \in \Theta$, let $P^\theta : \mathcal{S} \times \mathcal{S} \to [0, 1]$ denote the transition matrix of the Markov chain $\{s_t\}_{t \geq 0}$ induced by policy $\pi_\theta$, that is, for any $s, s' \in \mathcal{S}$, $P^\theta(s'|s) = \sum_{a \in \mathcal{A}} \pi_\theta(s, a) \cdot P(s'|s, a)$. We make the standard assumption that the Markov chain $\{s_t\}_{t \geq 0}$ is irreducible and aperiodic under any $\pi_\theta$ and denote its stationary distribution by $d_\theta$.

Our objective is to find a policy $\pi_\theta$ that maximizes the long-term average reward over the network. Let $r_{t+1}^i$ denote the reward received by agent $i$ as a result of taking action $a_t^i$. Then, we wish to solve:

$$\max_\theta J(\pi_\theta) = \lim_{T \to \infty} \frac{1}{T} \mathbb{E}\left[\sum_{t=0}^{T-1} \frac{1}{N} \sum_{i \in \mathcal{N}} r_{t+1}^i\right] = \sum_{s \in \mathcal{S}, a \in \mathcal{A}} d_\theta(s) \pi_\theta(s, a) \bar{R}(s, a),$$

where $\bar{R}(s, a) = (1/N) \cdot \sum_{i \in \mathcal{N}} R^i(s, a)$ is the globally averaged reward function. Let $\bar{r}_t = (1/N) \cdot \sum_{i \in \mathcal{N}} r_t^i$, then $\bar{R}(s, a) = \mathbb{E}[\bar{r}_{t+1}|s_t = s, a_t = a]$, and therefore, the global relative action-value function is: $Q_\theta(s, a) = \sum_{t \geq 0} \mathbb{E}[\bar{r}_{t+1} - J(\theta)|s_0 = s, a_0 = a, \pi_\theta]$, and the global relative state-value function is: $V_\theta(s) = \sum_{a \in \mathcal{A}} \pi_\theta(s, a) Q_\theta(s, a)$. For simplicity, we refer to $V_\theta$ and $Q_\theta$ as simply the state-value function and action-value function. We define the advantage function as $A_\theta(s, a) = Q_\theta(s, a) - V_\theta(s)$.

Zhang et al. [2018] provided the first provably convergent MARL algorithm in the context of the above model. The fundamental result underlying their algorithm is a local policy gradient theorem:

$$\nabla_{\theta^i} J(\mu_\theta) = \mathbb{E}_{s \sim d_\theta, a \sim \pi_\theta}\left[\nabla_{\theta^i} \log \pi_{\theta^i}^i(s, a^i) \cdot A_\theta^i(s, a)\right],$$

where $A_\theta^i(s, a) = Q_\theta(s, a) - \tilde{V}_\theta^i(s, a^{-i})$ is a local advantage function and $\tilde{V}_\theta^i(s, a^{-i}) = \sum_{a^i \in \mathcal{A}^i} \pi_{\theta^i}^i(s, a^i) Q_\theta(s, a^i, a^{-i})$. This theorem has important practical value as it shows that the policy gradient with respect to each local parameter $\theta^i$ can be obtained locally using the corresponding score function $\nabla_{\theta^i} \log \pi_{\theta^i}^i$ provided that agent $i$ has an unbiased estimate of the advantage functions $A_\theta^i$ or $A_\theta$. With only local information, the advantage functions $A_\theta^i$ or $A_\theta$ cannot be well estimated since the estimation requires the rewards $\{r_t^i\}_{i \in \mathcal{N}}$ of all agents. Therefore, they proposed a consensus based actor-critic that leverages the communication network to share information between agents by placing a weight $c_t(i, j)$ on the message transmitted from agent $j$ to agent $i$ at time $t$. Their action-value function $Q_\theta$ was approximated by a parameterized function $\hat{Q}_\omega : \mathcal{S} \times \mathcal{A} \to \mathbb{R}$, and each agent $i$ maintains its own parameter $\omega^i$, which it uses to form a local estimate $\hat{Q}_{\omega^i}$ of the global $Q_\theta$. At each time step $t$, each agent $i$ shares its local parameter $\omega_t^i$ with its neighbors on the network, and the shared parameters are used to arrive at a consensual estimate of $Q_\theta$ over time.

## 3 Local Gradients of Deterministic Policies

While the use of a stochastic policy facilitates the derivations of convergence proofs, most real-world control tasks require a deterministic policy to be implementable. In addition, the quantities estimated in the deterministic critic do not involve estimation of the complex integral over the action space found in the stochastic version. This offers lower variance of the critic estimates and faster convergence. To address the lack of exploration that comes with deterministic policies, we provide both off-policy and on-policy versions of our results. Our first requirement is a local deterministic policy gradient theorem.

We assume that $\mathcal{A}^i = \mathbb{R}^{n_i}$. We make standard regularity assumptions on our MDP. That is, we assume that for any $s, s' \in \mathcal{S}$, $P(s'|s, a)$ and $R^i(s, a)$ are bounded and have bounded first and second derivatives. We consider local deterministic policies $\mu^i_{\theta^i} : \mathcal{S} \to \mathcal{A}^i$ with parameter vector $\theta^i \in \Theta^i$, and denote the joint policy by $\mu_\theta : \mathcal{S} \to \mathcal{A}$, where $\mu_\theta(s) = (\mu^1_{\theta^1}(s), \ldots, \mu^N_{\theta^N}(s))$ and $\theta = [(\theta^1)^\top, \ldots, (\theta^N)^\top]^\top$. We assume that for any $s \in \mathcal{S}$, the deterministic policy function $\mu^i_{\theta^i}(s)$ is twice continuously differentiable with respect to the parameter $\theta^i$ over $\Theta^i$. Let $P^\theta$ denote the transition matrix of the Markov chain $\{s_t\}_{t \geq 0}$ induced by policy $\mu_\theta$, that is, for any $s, s' \in \mathcal{S}$, $P^\theta(s'|s) = P(s'|s, \mu_\theta(s))$. We assume that the Markov chain $\{s_t\}_{t \geq 0}$ is irreducible and aperiodic under any $\mu_\theta$ and denote its stationary distribution by $d^{\mu_\theta}$.

Our objective is to find a policy $\mu_\theta$ that maximizes the long-run average reward:

$$\max_\theta J(\mu_\theta) = \mathbb{E}_{s \sim d^{\mu_\theta}}[\bar{R}(s, \mu_\theta(s))] = \sum_{s \in \mathcal{S}} d^{\mu_\theta}(s)\bar{R}(s, \mu_\theta(s)).$$

Analogous to the stochastic policy case, we denote the action-value function by $Q_\theta(s, a) = \sum_{t \geq 0} \mathbb{E}[\bar{r}_{t+1} - J(\mu_\theta)|s_0 = s, a_0 = a, \mu_\theta]$, and the state-value function by $V_\theta(s) = Q_\theta(s, \mu_\theta(s))$. When there is no ambiguity, we will denote $J(\mu_\theta)$ and $d^{\mu_\theta}$ by simply $J(\theta)$ and $d^\theta$, respectively. We present three results for the long-run average reward: (1) an expression for the local deterministic policy gradient in the on-policy setting $\nabla_{\theta^i} J(\mu_\theta)$, (2) an expression for the gradient in the off-policy setting, and (3) we show that the deterministic policy gradient can be seen as the limit of the stochastic one.

**On-Policy Setting**

**Theorem 1** (Local Deterministic Policy Gradient Theorem - On Policy). *For any $\theta \in \Theta$, $i \in \mathcal{N}$, $\nabla_{\theta^i} J(\mu_\theta)$ exists and is given by*

$$\nabla_{\theta^i} J(\mu_\theta) = \mathbb{E}_{s \sim d^{\mu_\theta}} \left[ \nabla_{\theta^i} \mu^i_{\theta^i}(s) \nabla_{a^i} Q_\theta(s, \mu^{-i}_{\theta^{-i}}(s), a^i) \big|_{a^i = \mu^i_{\theta^i}(s)} \right].$$

The first step of the proof consists in showing that $\nabla_\theta J(\mu_\theta) = \mathbb{E}_{s \sim d^\theta} \left[ \nabla_\theta \mu_\theta(s) \nabla_a Q_\theta(s, a) \big|_{a = \mu_\theta(s)} \right]$. This is an extension of the well-known stochastic case, for which we have $\nabla_\theta J(\pi_\theta) = \mathbb{E}_{s \sim d_\theta} \left[ \nabla_\theta \log(\pi_\theta(a|s)) Q_\theta(s, a) \right]$, which holds for a long-term averaged return with stochastic policy (e.g Theorem 1 of Sutton et al. [2000a]). See the Appendix for the details.

**Off-Policy Setting** In the off-policy setting, we are given a behavior policy $\pi : \mathcal{S} \to \mathcal{P}(\mathcal{A})$, and our goal is to maximize the long-run average reward under state distribution $d^\pi$:

$$J_\pi(\mu_\theta) = \mathbb{E}_{s \sim d^\pi} \left[ \bar{R}(s, \mu_\theta(s)) \right] = \sum_{s \in \mathcal{S}} d^\pi(s)\bar{R}(s, \mu_\theta(s)). \tag{1}$$

Note that we consider here an excursion objective (Sutton et al. [2009], Silver et al. [January 2014a], Sutton et al. [2016]) since we take the average over the state distribution of the behaviour policy $\pi$ of the state-action reward when selecting action given by the target policy $\mu_\theta$. We thus have:

**Theorem 2** (Local Deterministic Policy Gradient Theorem - Off Policy). *For any $\theta \in \Theta$, $i \in \mathcal{N}$, $\pi : \mathcal{S} \to \mathcal{P}(\mathcal{A})$ a fixed stochastic policy, $\nabla_{\theta^i} J_\pi(\mu_\theta)$ exists and is given by*

$$\nabla_{\theta^i} J_\pi(\mu_\theta) = \mathbb{E}_{s \sim d^\pi} \left[ \nabla_{\theta^i} \mu^i_{\theta^i}(s) \nabla_{a^i} \bar{R}(s, \mu^{-i}_{\theta^{-i}}(s), a^i) \big|_{a^i = \mu^i_{\theta^i}(s)} \right].$$

117 *Proof.* Since $d^\pi$ is independent of $\theta$ we can take the gradient on both sides of (1)

$$\nabla_\theta J_\pi(\mu_\theta) = \mathbb{E}_{s \sim d^\pi} \left[ \nabla_\theta \mu_\theta(s) \, \nabla_a \bar{R}(s, \mu_\theta(s)) \big|_{a = \mu_\theta(s)} \right].$$

118 Given that $\nabla_{\theta^i} \mu_\theta^j(s) = 0$ if $i \neq j$, we have $\nabla_\theta \mu_\theta(s) = \mathrm{Diag}(\nabla_{\theta^1} \mu_{\theta_1}^1(s), \dots, \nabla_{\theta^N} \mu_{\theta_N}^N(s))$ and the
119 result follows. $\qquad\square$

120 This result implies that, off-policy, each agent needs access to $\mu_{\theta_t^{-i}}^{-i}(s_t)$ for every $t$.

121 **Limit Theorem**    As noted by Silver et al. [January 2014b], the fact that the deterministic gradient
122 is a limit case of the stochastic gradient enables the standard machinery of policy gradient, such as
123 compatible-function approximation (Sutton et al. [2000b]), natural gradients (Kakade [2001]), on-line
124 feature adaptation (Prabuchandran et al. [2016],) and actor-critic (Konda [2002]) to be used with
125 deterministic policies. We show that it holds in our setting. The proof can be found in the Appendix.

126 **Theorem 3** (Limit of the Stochastic Policy Gradient for MARL). *Let $\pi_{\theta,\sigma}$ be a stochastic policy*
127 *such that $\pi_{\theta,\sigma}(a|s) = \nu_\sigma(\mu_\theta(s), a)$, where $\sigma$ is a parameter controlling the variance, and $\nu_\sigma$ satisfy*
128 *Condition 1 in the Appendix. Then,*

$$\lim_{\sigma \downarrow 0} \nabla_\theta J_{\pi_{\theta,\sigma}}(\pi_{\theta,\sigma}) = \nabla_\theta J_{\mu_\theta}(\mu_\theta)$$

129 *where on the l.h.s the gradient is the standard stochastic policy gradient and on the r.h.s. the gradient*
130 *is the deterministic policy gradient.*

## 131 4    Algorithms

132 We provide two decentralized deterministic actor-critic algorithms, one on-policy and the other
133 off-policy and demonstrate their convergence in the next section; assumptions and proofs are provided
134 in the Appendix.

135 **On-Policy Deterministic Actor-Critic**

---
**Algorithm 1** Networked deterministic on-policy actor-critic

---
Initialize: step $t = 0$; parameters $\hat{J}_0^i, \omega_0^i, \widetilde{\omega}_0^i, \theta_0^i, \forall i \in \mathcal{N}$; state $s_0$; stepsizes $\{\beta_{\omega,t}\}_{t \geq 0}, \{\beta_{\theta,t}\}_{t \geq 0}$
Draw $a_0^i = \mu_{\theta_0^i}^i(s_0)$ and compute $\widetilde{a}_0^i = \nabla_{\theta^i} \mu_{\theta_0^i}^i(s_0)$
Observe joint action $a_0 = (a_0^1, \dots, a_0^N)$ and $\widetilde{a}_0 = (\widetilde{a}_0^1, \dots, \widetilde{a}_0^N)$
**repeat**
  **for** $i \in \mathcal{N}$ **do**
    Observe $s_{t+1}$ and reward $r_{t+1}^i = r^i(s_t, a_t)$
    Update $\hat{J}_{t+1}^i \leftarrow (1 - \beta_{\omega,t}) \cdot \hat{J}_t^i + \beta_{\omega,t} \cdot r_{t+1}^i$
    Draw action $a_{t+1}^i = \mu_{\theta_t^i}^i(s_{t+1})$ and compute $\widetilde{a}_{t+1}^i = \nabla_{\theta^i} \mu_{\theta_t^i}^i(s_{t+1})$
  **end for**
  Observe joint action $a_{t+1} = (a_{t+1}^1, \dots, a_{t+1}^N)$ and $\widetilde{a}_{t+1} = (\widetilde{a}_{t+1}^1, \dots, \widetilde{a}_{t+1}^N)$
  **for** $i \in \mathcal{N}$ **do**
    Update: $\delta_t^i \leftarrow r_{t+1}^i - \hat{J}_t^i + \hat{Q}_{\omega_t^i}(s_{t+1}, a_{t+1}) - \hat{Q}_{\omega_t^i}(s_t, a_t)$
    **Critic step:** $\widetilde{\omega}_t^i \leftarrow \omega_t^i + \beta_{\omega,t} \cdot \delta_t^i \cdot \nabla_\omega \hat{Q}_{\omega^i}(s_t, a_t) \big|_{\omega = \omega_t^i}$
    **Actor step:** $\theta_{t+1}^i = \theta_t^i + \beta_{\theta,t} \cdot \nabla_{\theta^i} \mu_{\theta_t^i}^i(s_t) \, \nabla_{a^i} \hat{Q}_{\omega_t^i}(s_t, a_t^{-i}, a^i) \big|_{a^i = a_t^i}$
    Send $\widetilde{\omega}_t^i$ to the neighbors $\{j \in \mathcal{N} : (i, j) \in \mathcal{E}_t\}$ over $\mathcal{G}_t$
    **Consensus step:** $\omega_{t+1}^i \leftarrow \sum_{j \in \mathcal{N}} c_t^{ij} \cdot \widetilde{\omega}_t^j$
  **end for**
  Update $t \leftarrow t + 1$
**until** end

---

Consider the following on-policy algorithm. The actor step is based on an expression for $\nabla_{\theta^i} J(\mu_\theta)$ in terms of $\nabla_{a^i} Q_\theta$ (see Equation (15) in the Appendix). We approximate the action-value function $Q_\theta$ using a family of functions $\hat{Q}_\omega : \mathcal{S} \times \mathcal{A} \to \mathbb{R}$ parameterized by $\omega$, a column vector in $\mathbb{R}^K$. Each agent $i$ maintains its own parameter $\omega^i$ and uses $\hat{Q}_{\omega^i}$ as its local estimate of $Q_\theta$. The parameters $\omega^i$ are updated in the critic step using consensus updates through a weight matrix $C_t = \left( c_t^{ij} \right)_{i,j} \in \mathbb{R}^{N \times N}$ where $c_t^{ij}$ is the weight on the message transmitted from $i$ to $j$ at time $t$, namely:

$$\hat{J}_{t+1}^i = (1 - \beta_{\omega,t}) \cdot \hat{J}_t^i + \beta_{\omega,t} \cdot r_{t+1}^i \tag{2}$$

$$\widetilde{\omega}_t^i = \omega_t^i + \beta_{\omega,t} \cdot \delta_t^i \cdot \nabla_\omega \hat{Q}_{\omega^i}(s_t, a_t)\Big|_{\omega=\omega_t^i} \tag{3}$$

$$\omega_{t+1}^i = \sum_{j \in \mathcal{N}} c_t^{ij} \cdot \widetilde{\omega}_t^j \tag{4}$$

with

$$\delta_t^i = r_{t+1}^i - \hat{J}_t^i + \hat{Q}_{\omega_t^i}(s_{t+1}, a_{t+1}) - \hat{Q}_{\omega_t^i}(s_t, a_t).$$

For the actor step, each agent $i$ improves its policy via:

$$\theta_{t+1}^i = \theta_t^i + \beta_{\theta,t} \cdot \nabla_{\theta^i} \mu_{\theta_t^i}^i(s_t) \cdot \nabla_{a^i} \hat{Q}_{\omega_t^i}(s_t, a_t^{-i}, a^i)\Big|_{a^i=a_t^i}. \tag{5}$$

Since Algorithm 1 is an on-policy algorithm, each agent updates the critic using only $(s_t, a_t, s_{t+1})$, at time $t$ knowing that $a_{t+1} = \mu_{\theta_t}(s_{t+1})$. The terms in blue are additional terms that need to be shared when using compatible features (this is explained further in the next section).

**Off-Policy Deterministic Actor-Critic** We further propose an off-policy actor-critic algorithm, defined in Algorithm 2 to enable better exploration capability. Here, the goal is to maximize $J_\pi(\mu_\theta)$ where $\pi$ is the behavior policy. To do so, the globally averaged reward function $\bar{R}(s, a)$ is approximated using a family of functions $\hat{\bar{R}}_\lambda : \mathcal{S} \times \mathcal{A} \to \mathbb{R}$ that are parameterized by $\lambda$, a column vector in $\mathbb{R}^K$. Each agent $i$ maintains its own parameter $\lambda^i$ and uses $\hat{\bar{R}}_{\lambda^i}$ as its local estimate of $\bar{R}$. Based on (1), the actor update is

$$\theta_{t+1}^i = \theta_t^i + \beta_{\theta,t} \cdot \nabla_{\theta^i} \mu_{\theta_t^i}^i(s_t) \cdot \nabla_{a^i} \hat{\bar{R}}_{\lambda_t^i}(s_t, \mu_{\theta_t^{-i}}^{-i}(s_t), a^i)\Big|_{a^i=\mu_{\theta_t^i}(s_t)}, \tag{6}$$

which requires each agent $i$ to have access to $\mu_{\theta_t^j}^j(s_t)$ for $j \in \mathcal{N}$.

The critic update is

$$\widetilde{\lambda}_t^i = \lambda_t^i + \beta_{\lambda,t} \cdot \delta_t^i \cdot \nabla_\lambda \hat{\bar{R}}_{\lambda^i}(s_t, a_t)\Big|_{\lambda=\lambda_t^i} \tag{7}$$

$$\lambda_{t+1}^i = \sum_{j \in \mathcal{N}} c_t^{ij} \widetilde{\lambda}_t^j, \tag{8}$$

with

$$\delta_t^i = r^i(s_t, a_t) - \hat{\bar{R}}_{\lambda_t^i}(s_t, a_t). \tag{9}$$

In this case, $\delta_t^i$ was motivated by distributed optimization results, and is not related to the local TD-error (as there is no "temporal" relationship for $R$). Rather, it is simply the difference between the sample reward and the bootstrap estimate. The terms in blue are additional terms that need to be shared when using compatible features (this is explained further in the next section).

# 5 Convergence

To show convergence, we use a two-timescale technique where in the actor, updating deterministic policy parameter $\theta^i$ occurs more slowly than that of $\omega^i$ and $\hat{J}^i$ in the critic. We study the asymptotic behaviour of the critic by freezing the joint policy $\mu_\theta$, then study the behaviour of $\theta_t$ under convergence of the critic. To ensure stability, projection is often assumed since it is not clear how boundedness of

---

**Algorithm 2** Networked deterministic off-policy actor-critic

---

Initialize: step $t = 0$; parameters $\lambda_0^i, \widetilde{\lambda}_0^i, \theta_0^i, \forall i \in \mathcal{N}$; state $s_0$; stepsizes $\{\beta_{\lambda,t}\}_{t \geq 0}, \{\beta_{\theta,t}\}_{t \geq 0}$

Draw $a_0^i \sim \pi^i(s_0)$, compute $\dot{a}_0^i = \mu_{\theta_0^i}^i(s_0)$ and $\widetilde{a}_0^i = \nabla_{\theta^i}\mu_{\theta_0^i}^i(s_0)$

Observe joint action $a_0 = (a_0^1, \ldots, a_0^N)$, $\dot{a}_0 = (\dot{a}_0^1, \ldots, \dot{a}_0^N)$ and $\widetilde{a}_0 = (\widetilde{a}_0^1, \ldots, \widetilde{a}_0^N)$

**repeat**

    **for** $i \in \mathcal{N}$ **do**

        Observe $s_{t+1}$ and reward $r_{t+1}^i = r^i(s_t, a_t)$

    **end for**

    **for** $i \in \mathcal{N}$ **do**

        Update: $\delta_t^i \leftarrow r_{t+1}^i - \hat{\bar{R}}_{\lambda_t^i}(s_t, a_t)$

        **Critic step:** $\widetilde{\lambda}_t^i \leftarrow \lambda_t^i + \beta_{\lambda,t} \cdot \delta_t^i \cdot \nabla_\lambda \hat{\bar{R}}_{\lambda^i}(s_t, a_t)\big|_{\lambda = \lambda_t^i}$

        **Actor step:** $\theta_{t+1}^i = \theta_t^i + \beta_{\theta,t} \cdot \nabla_{\theta^i}\mu_{\theta_t^i}^i(s_t) \cdot \nabla_{a^i}\hat{\bar{R}}_{\lambda_t^i}(s_t, \mu_{\theta_t^{-i}}^{-i}(s_t), a^i)\big|_{a^i = \mu_{\theta_t^i}(s_t)}$

        Send $\widetilde{\lambda}_t^i$ to the neighbors $\{j \in \mathcal{N} : (i,j) \in \mathcal{E}_t\}$ over $\mathcal{G}_t$

    **end for**

    **for** $i \in \mathcal{N}$ **do**

        **Consensus step:** $\lambda_{t+1}^i \leftarrow \sum_{j \in \mathcal{N}} c_t^{ij} \cdot \widetilde{\lambda}_t^j$

        Draw action $a_{t+1}^i \sim \pi(s_{t+1})$, compute $\dot{a}_{t+1}^i = \mu_{\theta_{t+1}^i}^i(s_{t+1})$ and compute $\widetilde{a}_{t+1}^i = \nabla_{\theta^i}\mu_{\theta_{t+1}^i}^i(s_{t+1})$

    **end for**

    Observe joint action $a_{t+1} = (a_{t+1}^1, \ldots, a_{t+1}^N)$, $\dot{a}_{t+1} = (\dot{a}_{t+1}^1, \ldots, \dot{a}_{t+1}^N)$ and $\widetilde{a}_{t+1} = (\widetilde{a}_{t+1}^1, \ldots, \widetilde{a}_{t+1}^N)$

    Update $t \leftarrow t + 1$

**until** end

---

$\{\theta_t^i\}$ can otherwise be ensured (see Bhatnagar et al. [2009]). However, in practice, convergence is typically observed even without the projection step (see Bhatnagar et al. [2009], Degris et al. [2012], Prabuchandran et al. [2016], Zhang et al. [2018], Suttle et al. [2019]). We also introduce the following technical assumptions which will be needed in the statement of the convergence results.

**Assumption 1** (Linear approximation, average-reward)**.** For each agent $i$, the average-reward function $\bar{R}$ is parameterized by the class of linear functions, i.e., $\hat{\bar{R}}_{\lambda^i,\theta}(s,a) = w_\theta(s,a) \cdot \lambda^i$ where $w_\theta(s,a) = [w_{\theta,1}(s,a), \ldots, w_{\theta,K}(s,a)] \in \mathbb{R}^K$ is the feature associated with the state-action pair $(s,a)$. The feature vectors $w_\theta(s,a)$, as well as $\nabla_a w_{\theta,k}(s,a)$ are uniformly bounded for any $s \in \mathcal{S}, a \in \mathcal{A}, k \in [\![1,K]\!]$. Furthermore, we assume that the feature matrix $W_\pi \in \mathbb{R}^{|S| \times K}$ has full column rank, where the $k$-th column of $W_{\pi,\theta}$ is $\left[\int_\mathcal{A} \pi(a|s)w_{\theta,k}(s,a)\mathrm{d}a, s \in \mathcal{S}\right]$ for any $k \in [\![1,K]\!]$.

**Assumption 2** (Linear approximation, action-value)**.** For each agent $i$, the action-value function is parameterized by the class of linear functions, i.e., $\hat{Q}_{\omega^i}(s,a) = \phi(s,a) \cdot \omega^i$ where $\phi(s,a) = [\phi_1(s,a), \ldots, \phi_K(s,a)] \in \mathbb{R}^K$ is the feature associated with the state-action pair $(s,a)$. The feature vectors $\phi(s,a)$, as well as $\nabla_a \phi_k(s,a)$ are uniformly bounded for any $s \in \mathcal{S}, a \in \mathcal{A}, k \in \{1, \ldots, K\}$. Furthermore, we assume that for any $\theta \in \Theta$, the feature matrix $\Phi_\theta \in \mathbb{R}^{|S| \times K}$ has full column rank, where the $k$-th column of $\Phi_\theta$ is $\left[\phi_k(s, \mu_\theta(s)), s \in \mathcal{S}\right]$ for any $k \in [\![1,K]\!]$. Also, for any $u \in \mathbb{R}^K$, $\Phi_\theta u \neq \mathbf{1}$.

**Assumption 3** (Bounding $\theta$)**.** The update of the policy parameter $\theta^i$ includes a local projection by $\Gamma^i : \mathbb{R}^{m_i} \to \Theta^i$ that projects any $\theta_t^i$ onto a compact set $\Theta^i$ that can be expressed as $\{\theta^i | q_j^i(\theta^i) \leq 0, j = 1, \ldots, s^i\} \subset \mathbb{R}^{m_i}$, for some real-valued, continuously differentiable functions $\{q_j^i\}_{1 \leq j \leq s^i}$ defined on $\mathbb{R}^{m_i}$. We also assume that $\Theta = \prod_{i=1}^N \Theta^i$ is large enough to include at least one local minimum of $J(\theta)$.

We use $\{\mathcal{F}_t\}$ to denote the filtration with $\mathcal{F}_t = \sigma(s_\tau, C_{\tau-1}, a_{\tau-1}, r_{\tau-1}, \tau \leq t)$.

**Assumption 4** (Random matrices)**.** The sequence of non-negative random matrices $\{C_t = (c_t^{ij})_{ij}\}$ satisfies:

1. $C_t$ is row stochastic and $\mathbb{E}(C_t|\mathcal{F}_t)$ is a.s. column stochastic for each $t$, i.e., $C_t\mathbf{1} = \mathbf{1}$ and $\mathbf{1}^\top \mathbb{E}(C_t|\mathcal{F}_t) = \mathbf{1}^\top$ a.s. Furthermore, there exists a constant $\eta \in (0,1)$ such that, for any $c_t^{ij} > 0$, we have $c_t^{ij} \geq \eta$.

2. $C_t$ respects the communication graph $\mathcal{G}_t$, i.e., $c_t^{ij} = 0$ if $(i,j) \notin \mathcal{E}_t$.

3. The spectral norm of $\mathbb{E}\big[C_t^\top \cdot (I - \mathbf{1}\mathbf{1}^\top/N) \cdot C_t\big]$ is smaller than one.

4. Given the $\sigma$-algebra generated by the random variables before time $t$, $C_t$, is conditionally independent of $s_t, a_t$ and $r_{t+1}^i$ for any $i \in \mathcal{N}$.

**Assumption 5** (Step size rules, on-policy). The stepsizes $\beta_{\omega,t}, \beta_{\theta,t}$ satisfy:

$$\sum_t \beta_{\omega,t} = \sum_t \beta_{\theta,t} = \infty$$

$$\sum_t (\beta_{\omega,t}^2 + \beta_{\theta,t}^2) < \infty$$

$$\sum_t |\beta_{\theta,t+1} - \beta_{\theta,t}| < \infty.$$

In addition, $\beta_{\theta,t} = o(\beta_{\omega,t})$ and $\lim_{t\to\infty} \beta_{\omega,t+1}/\beta_{\omega,t} = 1$.

**Assumption 6** (Step size rules, off-policy). The step-sizes $\beta_{\lambda,t}, \beta_{\theta,t}$ satisfy:

$$\sum_t \beta_{\lambda,t} = \sum_t \beta_{\theta,t} = \infty, \qquad \sum_t \beta_{\lambda,t}^2 + \beta_{\theta,t}^2 < \infty$$

$$\beta_{\theta,t} = o(\beta_{\lambda,t}), \qquad \lim_{t\to\infty} \beta_{\lambda,t+1}/\beta_{\lambda,t} = 1.$$

**On-Policy Convergence**   To state convergence of the critic step, we define $D_\theta^s = \text{Diag}\big[d^\theta(s), s \in \mathcal{S}\big]$, $\bar{R}_\theta = \big[\bar{R}(s, \mu_\theta(s)), s \in \mathcal{S}\big]^\top \in \mathbb{R}^{|\mathcal{S}|}$ and the operator $T_\theta^Q : \mathbb{R}^{|\mathcal{S}|} \to \mathbb{R}^{|\mathcal{S}|}$ for any action-value vector $Q \in \mathbb{R}^{|\mathcal{S}|}$ (and not $\mathbb{R}^{|\mathcal{S}|\cdot|\mathcal{A}|}$ since there is a mapping associating an action to each state) as:

$$T_\theta^Q(Q') = \bar{R}_\theta - J(\mu_\theta) \cdot \mathbf{1} + P^\theta Q'.$$

**Theorem 4.** *Under Assumptions 3, 4, and 5, for any given deterministic policy $\mu_\theta$, with $\{\hat{J}_t\}$ and $\{\omega_t\}$ generated from (2), we have $\lim_{t\to\infty} \frac{1}{N}\sum_{i\in\mathcal{N}} \hat{J}_t^i = J(\mu_\theta)$ and $\lim_{t\to\infty} \omega_t^i = \omega_\theta$ a.s. for any $i \in \mathcal{N}$, where*

$$J(\mu_\theta) = \sum_{s\in\mathcal{S}} d^\theta(s)\bar{R}(s, \mu_\theta(s))$$

*is the long-term average return under $\mu_\theta$, and $\omega_\theta$ is the unique solution to*

$$\Phi_\theta^\top D_\theta^s \big[T_\theta^Q(\Phi_\theta\omega_\theta) - \Phi_\theta\omega_\theta\big] = 0. \tag{10}$$

*Moreover, $\omega_\theta$ is the minimizer of the Mean Square Projected Bellman Error (MSPBE), i.e., the solution to*

$$\underset{\omega}{\text{minimize}} \; \|\Phi_\theta\omega - \Pi T_\theta^Q(\Phi_\theta\omega)\|_{D_\theta^s}^2,$$

*where $\Pi$ is the operator that projects a vector to the space spanned by the columns of $\Phi_\theta$, and $\|\cdot\|_{D_\theta^s}^2$ denotes the euclidean norm weighted by the matrix $D_\theta^s$.*

To state convergence of the actor step, we define quantities $\psi_{t,\theta}^i, \xi_t^i$ and $\xi_{t,\theta}^i$ as

$$\psi_{t,\theta}^i = \nabla_{\theta^i} \mu_{\theta^i}^i(s_t) \quad \text{and} \quad \psi_t^i = \psi_{t,\theta_t}^i = \nabla_{\theta^i} \mu_{\theta_t^i}^i(s_t),$$

$$\xi_{t,\theta}^i = \nabla_{a_i} \hat{Q}_{\omega_\theta}(s_t, a_t^{-i}, a_i)\Big|_{a_i = a_i = \mu_{\theta_t^i}^i(s_t)} = \nabla_{a_i} \phi(s_t, a_t^{-i}, a_i)\Big|_{a_i = a_i = \mu_{\theta_t^i}^i(s_t)} \omega_\theta,$$

$$\xi_t^i = \nabla_{a_i} \hat{Q}_{\omega_t^i}^i(s_t, a_t^{-i}, a_i)\Big|_{a_i = \mu_{\theta^i}^i(s_t)} = \nabla_{a_i} \phi(s_t, a_t^{-i}, a_i)\Big|_{a_i = \mu_{\theta^i}^i(s_t)} \omega_t^i.$$

Additionally, we introduce the operator $\hat{\Gamma}(\cdot)$ as

$$\hat{\Gamma}^i\left[g(\theta)\right] = \lim_{0<\eta\to 0} \frac{\Gamma^i\left[\theta^i + \eta \cdot g(\theta)\right] - \theta^i}{\eta} \tag{11}$$

for any $\theta \in \Theta$ and $g : \Theta \to \mathbb{R}^{m_i}$ a continuous function. In case the limit above is not unique we take $\hat{\Gamma}^i\left[g(\theta)\right]$ to be the set of all possible limit points of (11).

**Theorem 5.** *Under Assumptions 2, 3, 4, and 5, the policy parameter $\theta_t^i$ obtained from (5) converges a.s. to a point in the set of asymptotically stable equilibria of*

$$\dot{\theta}^i = \hat{\Gamma}^i\left[\mathbb{E}_{s_t \sim d^\theta, \mu_\theta}\left[\psi_{t,\theta}^i \cdot \xi_{t,\theta}^i\right]\right], \quad \text{for any } i \in \mathcal{N}. \tag{12}$$

*In the case of multiple limit points, the above is treated as a differential inclusion rather than an ODE.*

The convergence of the critic step can be proved by taking similar steps as that in Zhang et al. [2018]. For the convergence of the actor step, difficulties arise from the projection (which is handled using Kushner-Clark Lemma Kushner and Clark [1978]) and the state-dependent noise (that is handled by "natural" timescale averaging Crowder [2009]). Details are provided in the Appendix.

**Remark.** Note that that with a linear function approximator $Q_\theta$, $\psi_{t,\theta} \cdot \xi_{t,\theta} = \nabla_\theta \mu_\theta(s_t)\, \nabla_a \hat{Q}_{\omega_\theta}(s_t, a)\big|_{a=\mu_\theta(s_t)}$ may not be an unbiased estimate of $\nabla_\theta J(\theta)$:

$$\mathbb{E}_{s\sim d^\theta}\left[\psi_{t,\theta}\cdot\xi_{t,\theta}\right] = \nabla_\theta J(\theta) + \mathbb{E}_{s\sim d^\theta}\left[\nabla_\theta \mu_\theta(s) \cdot \left(\nabla_a \hat{Q}_{\omega_\theta}(s,a)\big|_{a=\mu_\theta(s)} - \nabla_a Q_{\omega_\theta}(s,a)\big|_{a=\mu_\theta(s)}\right)\right].$$

A standard approach to overcome this approximation issue is via compatible features (see, for example, Silver et al. [January 2014a] and Zhang and Zavlanos [2019]), i.e. $\phi(s,a) = a \cdot \nabla_\theta \mu_\theta(s)^\top$, giving, for $\omega \in \mathbb{R}^m$,

$$\hat{Q}_\omega(s,a) = a \cdot \nabla_\theta \mu_\theta(s)^\top \omega = (a - \mu_\theta(s)) \cdot \nabla_\theta \mu_\theta(s)^\top \omega + \hat{V}_\omega(s),$$

$$\text{with } \hat{V}_\omega(s) = \hat{Q}_\omega(s, \mu_\theta(s)) \quad \text{and} \quad \nabla_a \hat{Q}_\omega(s,a)\big|_{a=\mu_\theta(s)} = \nabla_\theta \mu_\theta(s)^\top \omega.$$

We thus expect that the convergent point of (5) corresponds to a small neighborhood of a local optimum of $J(\mu_\theta)$, i.e., $\nabla_{\theta^i} J(\mu_\theta) = 0$, provided that the error for the gradient of the action-value function $\nabla_a \hat{Q}_\omega(s,a)\big|_{a=\mu_\theta(s)} - \nabla_a Q_\theta(s,a)\big|_{a=\mu_\theta(s)}$ is small. However, note that using compatible features requires computing, at each step $t$, $\phi(s_t, a_t) = a_t \cdot \nabla_\theta \mu_\theta(s_t)^\top$. Thus, in Algorithm 1, each agent observes not only the joint action $a_{t+1} = (a_{t+1}^1, \ldots, a_{t+1}^N)$ but also $(\nabla_{\theta^1} \mu_{\theta_t^1}^1(s_{t+1}), \ldots, \nabla_{\theta^N} \mu_{\theta_t^N}^N(s_{t+1}))$ (see the parts in blue in Algorithm 1).

**Off-Policy Convergence**

**Theorem 6.** *Under Assumptions 1, 4, and 6, for any given behavior policy $\pi$ and any $\theta \in \Theta$, with $\{\lambda_t^i\}$ generated from (7), we have $\lim_{t\to\infty} \lambda_t^i = \lambda_\theta$ a.s. for any $i \in \mathcal{N}$, where $\lambda_\theta$ is the unique solution to*

$$B_{\pi,\theta} \cdot \lambda_\theta = A_{\pi,\theta} \cdot d_\pi^s \tag{13}$$

*where $d_\pi^s = \left[d^\pi(s), s \in \mathcal{S}\right]^\top$, $A_{\pi,\theta} = \left[\int_{\mathcal{A}} \pi(a|s)\bar{R}(s,a)w(s,a)^\top da, s \in \mathcal{S}\right] \in \mathbb{R}^{K\times|\mathcal{S}|}$ and $B_{\pi,\theta} = \left[\sum_{s\in\mathcal{S}} d^\pi(s) \int_{\mathcal{A}} \pi(a|s)w_i(s,a) \cdot w(s,a)^\top da, 1 \leq i \leq K\right] \in \mathbb{R}^{K\times K}$.*

From here on we let

$$\xi_{t,\theta}^i = \nabla_{a_i}\hat{\bar{R}}_{\lambda_\theta}(s_t, \mu_{\theta_t^{-i}}^{-i}(s_t), a_i)\big|_{a_i=\mu_{\theta_t^i}^i(s_t)} = \nabla_{a_i} w(s_t, \mu_{\theta_t^{-i}}^{-i}(s_t), a_i)\big|_{a_i=\mu_{\theta_t^i}^i(s_t)} \lambda_\theta$$

$$\xi_t^i = \nabla_{a_i}\hat{\bar{R}}_{\lambda_t^i}(s_t, \mu_{\theta_t^{-i}}^{-i}(s_t), a_i)\big|_{a_i=\mu_{\theta_t^i}^i(s_t)} = \nabla_{a_i} w(s_t, \mu_{\theta^{-i}}^{-i}(s_t), a_i)\big|_{a_i=\mu_{\theta_t^i}^i(s_t)} \lambda_t^i$$

and we keep

$$\psi_{t,\theta}^i = \nabla_{\theta^i} \mu_{\theta_t^i}^i(s_t), \quad \text{and} \quad \psi_t^i = \psi_{t,\theta_t}^i = \nabla_{\theta^i} \mu_{\theta_t^i}^i(s_t).$$

**Theorem 7.** *Under Assumptions 1, 3, 4, and 6, the policy parameter $\theta_t^i$ obtained from (6) converges a.s. to a point in the asymptotically stable equilibria of*

$$\dot{\theta}^i = \Gamma^i \left[ \mathbb{E}_{s \sim d^\pi} \left[ \psi_{t,\theta}^i \cdot \xi_{t,\theta}^i \right] \right]. \tag{14}$$

We define compatible features for the action-value and the average-reward function in an analogous manner: $w_\theta(s,a) = (a - \mu_\theta(s)) \cdot \nabla_\theta \mu_\theta(s)^\top$. For $\lambda \in \mathbb{R}^m$,

$$\hat{\bar{R}}_{\lambda,\theta}(s,a) = (a - \mu_\theta(s)) \cdot \nabla_\theta \mu_\theta(s)^\top \cdot \lambda$$

$$\nabla_a \hat{\bar{R}}_{\lambda,\theta}(s,a) = \nabla_\theta \mu_\theta(s)^\top \cdot \lambda$$

and we have that, for $\lambda^* = \underset{\lambda}{\operatorname{argmin}} \ \mathbb{E}_{s \sim d^\pi} \left[ \| \nabla_a \hat{\bar{R}}_{\lambda,\theta}(s, \mu_\theta(s)) - \nabla_a \bar{R}(s, \mu_\theta(s)) \|^2 \right]$:

$$\nabla_\theta J_\pi(\mu_\theta) = \mathbb{E}_{s \sim d^\pi} \left[ \nabla_\theta \mu_\theta(s) \cdot \nabla_a \bar{R}(s,a) \big|_{a = \mu_\theta(s)} \right] = \mathbb{E}_{s \sim d^\pi} \left[ \nabla_\theta \mu_\theta(s) \cdot \nabla_a \hat{\bar{R}}_{\lambda^*,\theta}(s,a) \big|_{a = \mu_\theta(s)} \right].$$

The use of compatible features requires each agent to observe not only the joint action taken $a_{t+1} = (a_{t+1}^1, \ldots, a_{t+1}^N)$ and the "on-policy action" $\dot{a}_{t+1} = (\dot{a}_{t+1}^1, \ldots, \dot{a}_{t+1}^N)$, but also $\tilde{a}_{t+1} = (\nabla_{\theta^1} \mu_{\theta_t^1}^1(s_{t+1}), \ldots, \nabla_{\theta^N} \mu_{\theta_t^N}^N(s_{t+1}))$ (see the parts in blue in Algorithm 2).

We illustrate algorithm convergence on multi-agent extension of a continuous bandit problem from Sec. 5.1 of Silver et al. [January 2014b]. Details are in the Appendix. Figure 2 shows the convergence of Algorithms 1 and 2 averaged over 5 runs. In all cases, the system converges and the agents are able to coordinate their actions to minimize system cost.

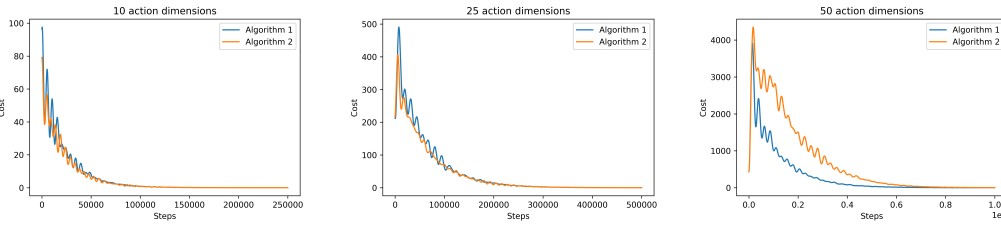

Figure 1: Convergence of Algorithms 1 and 2 on the multi-agent continuous bandit problem.

## 6   Conclusion

We have provided the tools needed to implement decentralized, deterministic actor-critic algorithms for cooperative multi-agent reinforcement learning. We provide the expressions for the policy gradients, the algorithms themselves, and prove their convergence in on-policy and off-policy settings. We also provide numerical results for a continuous multi-agent bandit problem that demonstrates the convergence of our algorithms. Our work differs from Zhang and Zavlanos [2019] as the latter was based on policy consensus whereas ours is based on critic consensus. Our approach represents agreement between agents on every participants' contributions to the global reward, and as such, provides a consensus scoring function with which to evaluate agents. Our approach may be used in compensation schemes to incentivize participation. An interesting extension of this work would be to prove convergence of our actor-critic algorithm for continuous state spaces, as it may hold with assumptions on the geometric ergodicity of the stationary state distribution induced by the deterministic policies (see Crowder [2009]). The expected policy gradient (EPG) of Ciosek and Whiteson [2018], a hybrid between stochastic and deterministic policy gradient, would also be interesting to leverage. The Multi-Agent Deep Deterministic Policy Gradient algorithm (MADDPG) of Lowe et al. [2017] assumes partial observability for each agent and would be a useful extension, but it is likely difficult to extend our convergence guarantees to the partially observed setting.

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

## Numerical experiment details

We demonstrate the convergence of our algorithm in a continuous bandit problem that is a multi-agent extension of the experiment in Section 5.1 of Silver et al. (2014). Each agent chooses an action $a^i \in \mathbb{R}^m$. We assume all agents have the same reward function given by $R^i(a) = -\left(\sum_i a^i - a^*\right)^\mathsf{T} C \left(\sum_i a^i - a^*\right)$. The matrix $C$ is positive definite with eigenvalues chosen from $\{0.1, 1\}$, and $a^* = [4, \ldots, 4]^\mathsf{T}$. We consider $10$ agents and action dimensions $m = 10, 20, 50$. Note that there are multiple possible solutions for this problem, requiring the agents to coordinate their actions to sum to $a^*$. We assume a target policy of the form $\mu_{\theta^i} = \theta^i$ for each agent $i$ and a Gaussian behaviour policy $\beta(\cdot) \sim \mathcal{N}(\theta^i, \sigma_\beta^2)$ where $\sigma_\beta = 0.1$. We use the Gaussian behaviour policy for both Algorithms 1 and 2. Strictly speaking, Algorithm 1 is on-policy, but in this simplified setting where the target policy is constant, the on-policy version would be degenerate such that the $Q$ estimate does not affect the TD-error. Therefore, we add a Gaussian behaviour policy to Algorithm 1. Each agent maintains an estimate $Q^{\omega^i}(a)$ of the critic using a linear function of the compatible features $a - \theta$ and a bias feature. The critic is recomputed from each successive batch of $2m$ steps and the actor is updated once per batch. The critic step size is $0.1$ and the actor step size is $0.01$. Performance is evaluated by measuring the cost of the target policy (without exploration). Figure 2 shows the convergence of Algorithms 1 and 2 averaged over 5 runs. In all cases, the system converges and the agents are able to coordinate their actions to minimize system cost. The jupyter notebook will be made available for others to use. In fact, in this simple experiment, we also observe convergence under discounted rewards.

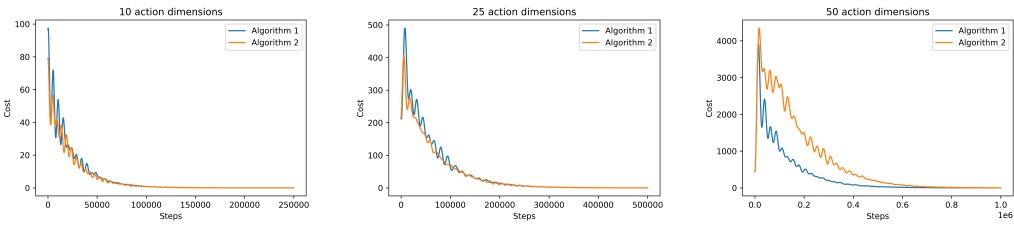

Figure 2: Convergence of Algorithms 1 and 2 on the multi-agent continuous bandit problem.

## Proof of Theorem 1

The proof follows the same scheme as Sutton et al. [2000a], naturally extending their results for a deterministic policy $\mu_\theta$ and a continuous action space $\mathcal{A}$.

Note that our regularity assumptions ensure that, for any $s \in \mathcal{S}$, $V_\theta(s)$, $\nabla_\theta V_\theta(s)$, $J(\theta)$, $\nabla_\theta J(\theta)$, $d^\theta(s)$ are Lipschitz-continuous functions of $\theta$ (since $\mu_\theta$ is twice continuously differentiable and $\Theta$ is compact), and that $Q_\theta(s, a)$ and $\nabla_a Q_\theta(s, a)$ are Lipschitz-continuous functions of $a$ (Marbach and Tsitsiklis [2001]).

We first show that $\nabla_\theta J(\theta) = \mathbb{E}_{s \sim d^\theta}\left[\nabla_\theta \mu_\theta(s) \nabla_a Q_\theta(s, a)|_{a = \mu_\theta(s)}\right]$.

The Poisson equation under policy $\mu_\theta$ is given by Puterman [1994]

$$Q_\theta(s, a) = \bar{R}(s, a) - J(\theta) + \sum_{s' \in \mathcal{S}} P(s'|s, a) V_\theta(s').$$

358 So,

$$\nabla_\theta V_\theta(s) = \nabla_\theta Q_\theta(s, \mu_\theta(s))$$

$$= \nabla_\theta \big[ \bar{R}(s, \mu_\theta(s)) - J(\theta) + \sum_{s' \in \mathcal{S}} P(s'|s, \mu_\theta(s)) V_\theta(s') \big]$$

$$= \nabla_\theta \mu_\theta(s) \left. \nabla_a \bar{R}(s, a) \right|_{a = \mu_\theta(s)} - \nabla_\theta J(\theta) + \nabla_\theta \sum_{s' \in \mathcal{S}} P(s'|s, \mu_\theta(s)) V_\theta(s')$$

$$= \nabla_\theta \mu_\theta(s) \left. \nabla_a \bar{R}(s, a) \right|_{a = \mu_\theta(s)} - \nabla_\theta J(\theta)$$

$$\quad + \sum_{s' \in \mathcal{S}} \nabla_\theta \mu_\theta(s) \left. \nabla_a P(s'|s, a) \right|_{a = \mu_\theta(s)} V_\theta(s') + \sum_{s' \in \mathcal{S}} P(s'|s, \mu_\theta(s)) \nabla_\theta V_\theta(s')$$

$$= \nabla_\theta \mu_\theta(s) \nabla_a \left[ \bar{R}(s, a) + \sum_{s' \in \mathcal{S}} P(s|s', a) V_\theta(s') \right] \Bigg|_{a = \mu_\theta(s)}$$

$$\quad - \nabla_\theta J(\theta) + \sum_{s' \in \mathcal{S}} P(s'|s, \mu_\theta(s)) \nabla_\theta V_\theta(s')$$

$$= \nabla_\theta \mu_\theta(s) \left. \nabla_a Q_\theta(s, a) \right|_{a = \mu_\theta(s)} + \sum_{s' \in \mathcal{S}} P(s'|s, \mu_\theta(s)) \nabla_\theta V_\theta(s') - \nabla_\theta J(\theta)$$

359 Hence,

$$\nabla_\theta J(\theta) = \nabla_\theta \mu_\theta(s) \left. \nabla_a Q_\theta(s, a) \right|_{a = \mu_\theta(s)} + \sum_{s' \in \mathcal{S}} P(s'|s, \mu_\theta(s)) \nabla_\theta V_\theta(s') - \nabla_\theta V_\theta(s)$$

$$\sum_{s \in \mathcal{S}} d^\theta(s) \nabla_\theta J(\theta) = \sum_{s \in \mathcal{S}} d^\theta(s) \nabla_\theta \mu_\theta(s) \left. \nabla_a Q_\theta(s, a) \right|_{a = \mu_\theta(s)}$$

$$\quad + \sum_{s \in \mathcal{S}} d^\theta(s) \sum_{s' \in \mathcal{S}} P(s'|s, \mu_\theta(s)) \nabla_\theta V_\theta(s') - \sum_{s \in \mathcal{S}} d^\theta(s) \nabla_\theta V_\theta(s).$$

Using stationarity property of $d^\theta$, we get

$$\sum_{s \in \mathcal{S}} \sum_{s' \in \mathcal{S}} d^\theta(s) P(s'|s, \mu_\theta(s)) \nabla_\theta V_\theta(s') = \sum_{s' \in \mathcal{S}} d^\theta(s') \nabla_\theta V_\theta(s').$$

Therefore, we get

$$\nabla_\theta J(\theta) = \sum_{s \in \mathcal{S}} d^\theta(s) \nabla_\theta \mu_\theta(s) \left. \nabla_a Q_\theta(s, a) \right|_{a = \mu_\theta(s)} = \mathbb{E}_{s \sim d^\theta} \big[ \nabla_\theta \mu_\theta(s) \left. \nabla_a Q_\theta(s, a) \right|_{a = \mu_\theta(s)} \big].$$

360 Given that $\nabla_{\theta^i} \mu_\theta^j(s) = 0$ if $i \neq j$, we have $\nabla_\theta \mu_\theta(s) = \mathrm{Diag}(\nabla_{\theta^1} \mu_{\theta_1}^1(s), \dots, \nabla_{\theta^N} \mu_{\theta_N}^N(s))$, which
361 implies

$$\nabla_{\theta^i} J(\theta) = \mathbb{E}_{s \sim d^\theta} \big[ \nabla_{\theta^i} \mu_{\theta^i}^i(s) \left. \nabla_{a^i} Q_\theta(s, \mu_{\theta^{-i}}^{-i}(s), a^i) \right|_{a^i = \mu_{\theta^i}^i(s)} \big]. \tag{15}$$

362 **Proof of Theorem 3**

363 We extend the notation for off-policy reward function to stochastic policies as follows. Let $\beta$ be a
364 behavior policy under which $\{s_t\}_{t \geq 0}$ is irreducible and aperiodic, with stationary distribution $d^\beta$. For
365 a stochastic policy $\pi : \mathcal{S} \to \mathcal{P}(\mathcal{A})$, we define

$$J_\beta(\pi) = \sum_{s \in \mathcal{S}} d^\beta(s) \int_{\mathcal{A}} \pi(a|s) \bar{R}(s, a) \mathrm{d}a.$$

366 Recall that for a deterministic policy $\mu : \mathcal{S} \to \mathcal{A}$, we have

$$J_\beta(\mu) = \sum_{s \in \mathcal{S}} d^\beta(s) \bar{R}(s, \mu(s)).$$

367 We introduce the following conditions which are identical to **Conditions B1** from Silver et al.
368 [January 2014a].

**Conditions 1.** Functions $\nu_\sigma$ parametrized by $\sigma$ are said to be regular delta-approximation on $\mathcal{R} \subset \mathcal{A}$ if they satisfy the following conditions:

1. The distributions $\nu_\sigma$ converge to a delta distribution: $\lim_{\sigma \downarrow 0} \int_\mathcal{A} \nu_\sigma(a', a) f(a) \mathrm{d}a = f(a')$ for $a' \in \mathcal{R}$ and suitably smooth $f$. Specifically we require that this convergence is uniform in $a'$ and over any class $\mathcal{F}$ of $L$-Lipschitz and bounded functions, $\|\nabla_a f(a)\| < L < \infty$, $\sup_a f(a) < b < \infty$, i.e.:

$$\lim_{\sigma \downarrow 0} \sup_{f \in \mathcal{F}, a' \in \mathcal{R}} \left| \int_\mathcal{A} \nu_\sigma(a', a) f(a) \mathrm{d}a - f(a') \right| = 0.$$

2. For each $a' \in \mathcal{R}$, $\nu_\sigma(a', \cdot)$ is supported on some compact $\mathcal{C}_{a'} \subseteq \mathcal{A}$ with Lipschitz boundary $\mathrm{bd}(\mathcal{C}_{a'})$, vanishes on the boundary and is continuously differentiable on $\mathcal{C}_{a'}$.

3. For each $a' \in \mathcal{R}$, for each $a \in \mathcal{A}$, the gradient $\nabla_{a'} \nu_\sigma(a', a)$ exists.

4. Translation invariance: for all $a \in \mathcal{A}, a' \in \mathcal{R}$, and any $\delta \in \mathbb{R}^n$ such that $a + \delta \in \mathcal{A}$, $a' + \delta \in \mathcal{A}$, $\nu_\sigma(a', a) = \nu_\sigma(a' + \delta, a + \delta)$.

The following lemma is an immediate corollary of **Lemma 1** from Silver et al. [January 2014a].

**Lemma 1.** *Let $\nu_\sigma$ be a regular delta-approximation on $\mathcal{R} \subseteq \mathcal{A}$. Then, wherever the gradients exist*

$$\nabla_{a'} \nu(a', a) = -\nabla_a \nu(a', a).$$

Theorem 3 is a less technical restatement of the following result.

**Theorem 8.** *Let $\mu_\theta : \mathcal{S} \to \mathcal{A}$. Denote the range of $\mu_\theta$ by $\mathcal{R}_\theta \subseteq \mathcal{A}$, and $\mathcal{R} = \cup_\theta \mathcal{R}_\theta$. For each $\theta$, consider $\pi_{\theta,\sigma}$ a stochastic policy such that $\pi_{\theta,\sigma}(a|s) = \nu_\sigma(\mu_\theta(s), a)$, where $\nu_\sigma$ satisfy Conditions 1 on $\mathcal{R}$. Then, there exists $r > 0$ such that, for each $\theta \in \Theta$, $\sigma \mapsto J_{\pi_{\theta,\sigma}}(\pi_{\theta,\sigma})$, $\sigma \mapsto J_{\pi_{\theta,\sigma}}(\mu_\theta)$, $\sigma \mapsto \nabla_\theta J_{\pi_{\theta,\sigma}}(\pi_{\theta,\sigma})$, and $\sigma \mapsto \nabla_\theta J_{\pi_{\theta,\sigma}}(\mu_\theta)$ are properly defined on $[0, r]$ (with $J_{\pi_{\theta,0}}(\pi_{\theta,0}) = J_{\pi_{\theta,0}}(\mu_\theta) = J_{\mu_\theta}(\mu_\theta)$ and $\nabla_\theta J_{\pi_{\theta,0}}(\pi_{\theta,0}) = \nabla_\theta J_{\pi_{\theta,0}}(\mu_\theta) = \nabla_\theta J_{\mu_\theta}(\mu_\theta)$), and we have:*

$$\lim_{\sigma \downarrow 0} \nabla_\theta J_{\pi_{\theta,\sigma}}(\pi_{\theta,\sigma}) = \lim_{\sigma \downarrow 0} \nabla_\theta J_{\pi_{\theta,\sigma}}(\mu_\theta) = \nabla_\theta J_{\mu_\theta}(\mu_\theta).$$

To prove this result, we first state and prove the following Lemma.

**Lemma 2.** *There exists $r > 0$ such that, for all $\theta \in \Theta$ and $\sigma \in [0, r]$, stationary distribution $d^{\pi_{\theta,\sigma}}$ exists and is unique. Moreover, for each $\theta \in \Theta$, $\sigma \mapsto d^{\pi_{\theta,\sigma}}$ and $\sigma \mapsto \nabla_\theta d^{\pi_{\theta,\sigma}}$ are properly defined on $[0, r]$ and both are continuous at $0$.*

*Proof of Lemma 2.* For any policy $\beta$, we let $\left( P^\beta_{s,s'} \right)_{s,s' \in \mathcal{S}}$ be the transition matrix associated to the Markov Chain $\{s_t\}_{t \geq 0}$ induced by $\beta$. In particular, for each $\theta \in \Theta$, $\sigma > 0$, $s, s' \in \mathcal{S}$, we have

$$P^{\mu_\theta}_{s,s'} = P(s'|s, \mu_\theta(s)),$$

$$P^{\pi_{\theta,\sigma}}_{s,s'} = \int_\mathcal{A} \pi_{\theta,\sigma}(a|s) P(s'|s, a) \mathrm{d}a = \int_\mathcal{A} \nu_\sigma(\mu_\theta(s), a) P(s'|s, a) \mathrm{d}a.$$

Let $\theta \in \Theta$, $s, s' \in \mathcal{S}$, $(\theta_n) \in \Theta^\mathbb{N}$ such that $\theta_n \to \theta$ and $(\sigma_n)_{n \in \mathbb{N}} \in \mathbb{R}^{+\mathbb{N}}$, $\sigma_n \downarrow 0$:

$$\left| P^{\pi_{\theta_n,\sigma_n}}_{s,s'} - P^{\mu_\theta}_{s,s'} \right| \leq \left| P^{\pi_{\theta_n,\sigma_n}}_{s,s'} - P^{\mu_{\theta_n}}_{s,s'} \right| + \left| P^{\mu_{\theta_n}}_{s,s'} - P^{\mu_\theta}_{s,s'} \right|.$$

Applying the first condition of Conditions 1 with $f : a \mapsto P(s'|s, a)$ belonging to $\mathcal{F}$:

$$\left| P^{\pi_{\theta_n,\sigma_n}}_{s,s'} - P^{\mu_{\theta_n}}_{s,s'} \right| = \left| \int_\mathcal{A} \nu_{\sigma_n}(\mu_{\theta_n}(s), a) P(s'|s, a) \mathrm{d}a - P(s'|s, \mu_{\theta_n}(s)) \right|$$

$$\leq \sup_{f \in \mathcal{F}, a' \in \mathcal{R}} \left| \int_\mathcal{A} \nu_{\sigma_n}(a', a) f(a) \mathrm{d}a - f(a') \right| \underset{n \to \infty}{\longrightarrow} 0.$$

By regularity assumptions on $\theta \mapsto \mu_\theta(s)$ and $P(s'|s, \cdot)$, we have

$$\left| P^{\mu_{\theta_n}}_{s,s'} - P^{\mu_\theta}_{s,s'} \right| = |P(s'|s, \mu_{\theta_n}(s)) - P(s'|s, \mu_\theta(s))| \underset{n \to \infty}{\longrightarrow} 0.$$

397 Hence,

$$\left| P^{\pi_{\theta_n},\sigma_n}_{s,s'} - P^{\mu_\theta}_{s,s'} \right| \underset{n\to\infty}{\longrightarrow} 0.$$

398 Therefore, for each $s, s' \in \mathcal{S}$, $(\theta, \sigma) \mapsto P^{\pi_\theta,\sigma}_{s,s'}$, with $P^{\pi_\theta,0}_{s,s'} = P^{\mu_\theta}_{s,s'}$, is continuous on $\Theta \times \{0\}$. Note
399 that, for each $n \in \mathbb{N}$, $P \mapsto \prod_{s,s'} (P^n)_{s,s'}$ is a polynomial function of the entries of $P$. Thus, for
400 each $n \in \mathbb{N}$, $f_n : (\theta, \sigma) \mapsto \prod_{s,s'} \left( P^{\pi_\theta,\sigma\, n} \right)_{s,s'}$, with $f_n(\theta, 0) = \prod_{s,s'} \left( P^{\mu_\theta\, n} \right)_{s,s'}$ is continuous on
401 $\Theta \times \{0\}$. Moreover, for each $\theta \in \Theta, \sigma \geq 0$, from the structure of $P^{\pi_\theta,\sigma}$, if there is some $n^* \in \mathbb{N}$
402 such that $f_{n^*}(\theta, \sigma) > 0$ then, for all $n \geq n^*$, $f_n(\theta, \sigma) > 0$.

403 Now let us suppose that there exists $(\theta_n) \in \Theta^{\mathbb{N}^*}$ such that, for each $n > 0$ there is a $\sigma_n \leq n^{-1}$ such
404 that $f_n(\theta_n, \sigma_n) = 0$. By compacity of $\Theta$, we can take $(\theta_n)$ converging to some $\theta \in \Theta$. For each
405 $n^* \in \mathbb{N}$, by continuity we have $f_{n^*}(\theta, 0) = \lim\limits_{n\to\infty} f_{n^*}(\theta_n, \sigma_n) = 0$. Since $P^{\mu_\theta}$ is irreducible and

406 aperiodic, there is some $n \in \mathbb{N}$ such that for all $s, s' \in \mathcal{S}$ and for all $n^* \geq n$, $\left( P^{\mu_\theta\, n^*} \right)_{s,s'} > 0$, i.e.

407 $f_{n^*}(\theta, 0) > 0$. This leads to a contradiction.

408 Hence, there exists $n^* > 0$ such that for all $\theta \in \Theta$ and $\sigma \leq n^{*-1}$, $f_n(\theta, \sigma) > 0$. We let $r = n^{*-1}$. It
409 follows that, for all $\theta \in \Theta$ and $\sigma \in [0, r]$, $P^{\pi_\theta,\sigma}$ is a transition matrix associated to an irreducible and
410 aperiodic Markov Chain, thus $d^{\pi_\theta,\sigma}$ is well defined as the unique stationary probability distribution
411 associated to $P^{\pi_\theta,\sigma}$. We fix $\theta \in \Theta$ in the remaining of the proof.

412 Let $\beta$ a policy for which the Markov Chain corresponding to $P^\beta$ is irreducible and aperiodic. Let
413 $s_* \in \mathcal{S}$, as asserted in Marbach and Tsitsiklis [2001], considering stationary distribution $d^\beta$ as a
414 vector $\left( d^\beta_s \right)_{s\in\mathcal{S}} \in \mathbb{R}^{|\mathcal{S}|}$, $d^\beta$ is the unique solution of the balance equations:

$$\sum_{s\in\mathcal{S}} d^\beta_s P^\beta_{s,s'} = d^\beta_{s'}, \quad s' \in \mathcal{S}\backslash\{s_*\},$$

$$\sum_{s\in\mathcal{S}} d^\beta_s = 1.$$

415 Hence, we have $A^\beta$ an $|\mathcal{S}| \times |\mathcal{S}|$ matrix and $a \neq 0$ a constant vector of $\mathbb{R}^{|\mathcal{S}|}$ such that the balance
416 equations is of the form

$$A^\beta d^\beta = a \tag{16}$$

417 with $A^\beta_{s,s'}$ depending on $P^\beta_{s',s}$ in an affine way, for each $s, s' \in \mathcal{S}$. Moreover, $A^\beta$ is invertible, thus
418 $d^\beta$ is given by

$$d^\beta = \frac{1}{\det(A^\beta)} \mathrm{adj}(A^\beta)^\top a.$$

419 Entries of $\mathrm{adj}(A^\beta)$ and $\det(A^\beta)$ are polynomial functions of the entries of $P^\beta$.

420 Thus, $\sigma \mapsto d^{\pi_\theta,\sigma} = \frac{1}{\det(A^{\pi_\theta,\sigma})} \mathrm{adj}(A^{\pi_\theta,\sigma})^\top a$ is defined on $[0, r]$ and is continuous at 0.

421 Lemma 1 and integration by parts imply that, for $s, s' \in \mathcal{S}, \sigma \in [0, r]$:

$$\int_{\mathcal{A}} \nabla_{a'}\nu_\sigma(a', a)|_{a'=\mu_\theta(s)} P(s'|s, a)\mathrm{d}a = -\int_{\mathcal{A}} \nabla_a\nu_\sigma(\mu_\theta(s), a)P(s'|s, a)\mathrm{d}a$$

$$= \int_{\mathcal{C}_{\mu_\theta(s)}} \nu_\sigma(\mu_\theta(s), a)\nabla_a P(s'|s, a)\mathrm{d}a + \text{boundary terms}$$

$$= \int_{\mathcal{C}_{\mu_\theta(s)}} \nu_\sigma(\mu_\theta(s), a)\nabla_a P(s'|s, a)\mathrm{d}a$$

422 where the boundary terms are zero since $\nu_\sigma$ vanishes on the boundary due to Conditions 1.

423 Thus, for $s, s' \in \mathcal{S}$, $\sigma \in [0, r]$:

$$\nabla_\theta P_{s,s'}^{\pi_{\theta,\sigma}} = \nabla_\theta \int_{\mathcal{A}} \pi_{\theta,\sigma}(a|s) P(s'|s,a) \mathrm{d}a$$

$$= \int_{\mathcal{A}} \nabla_\theta \pi_{\theta,\sigma}(a|s) P(s'|s,a) \mathrm{d}a \qquad (17)$$

$$= \int_{\mathcal{A}} \nabla_\theta \mu_\theta(s) \, \nabla_{a'} \nu_\sigma(a', a)|_{a'=\mu_\theta(s)} P(s'|s,a) \mathrm{d}a$$

$$= \nabla_\theta \mu_\theta(s) \int_{\mathcal{C}_{\mu_\theta(s)}} \nu_\sigma(\mu_\theta(s), a) \nabla_a P(s'|s,a) \mathrm{d}a$$

424 where exchange of derivation and integral in (17) follows by application of Leibniz rule with:

425 • $\forall a \in \mathcal{A}$, $\theta \mapsto \pi_{\theta,\sigma}(a|s) P(s'|s,a)$ is differentiable, and $\nabla_\theta \pi_{\theta,\sigma}(a|s) P(s'|s,a) =$
426 $\nabla_\theta \mu_\theta(s) \, \nabla_{a'} \nu_\sigma(a', a)|_{a'=\mu_\theta(s)}$.
427

428 • Let $a^* \in \mathcal{R}$, $\forall \theta \in \Theta$,

$$\|\nabla_\theta \pi_{\theta,\sigma}(a|s) P(s'|s,a)\| = \left\| \nabla_\theta \mu_\theta(s) \, \nabla_{a'} \nu_\sigma(a', a)|_{a'=\mu_\theta(s)} \right\|$$

$$\leq \|\nabla_\theta \mu_\theta(s)\|_{\mathrm{op}} \left\| \nabla_{a'} \nu_\sigma(a', a)|_{a'=\mu_\theta(s)} \right\|$$

$$\leq \sup_{\theta \in \Theta} \|\nabla_\theta \mu_\theta(s)\|_{\mathrm{op}} \|\nabla_a \nu_\sigma(\mu_\theta(s), a)\|$$

$$= \sup_{\theta \in \Theta} \|\nabla_\theta \mu_\theta(s)\|_{\mathrm{op}} \|\nabla_a \nu_\sigma(a^*, a - \mu_\theta(s) + a^*)\| \qquad (18)$$

$$\leq \sup_{\theta \in \Theta} \|\nabla_\theta \mu_\theta(s)\|_{\mathrm{op}} \sup_{a \in \mathcal{C}_{a^*}} \|\nabla_a \nu_\sigma(a^*, a)\| \, \mathbf{1}_{a \in \mathcal{C}_{a^*}}$$

429 where $\|\cdot\|_{\mathrm{op}}$ denotes the operator norm, and (18) comes from translation invariance (we take
430 $\nabla_a \nu_\sigma(a^*, a) = 0$ for $a \in \mathbb{R}^n \backslash \mathcal{C}_{a^*}$). $a \mapsto \sup_{\theta \in \Theta} \|\nabla_\theta \mu_\theta(s)\|_{\mathrm{op}} \sup_{a \in \mathcal{C}_{a^*}} \|\nabla_a \nu_\sigma(a^*, a)\| \, \mathbf{1}_{a \in \mathcal{C}_{a^*}}$ is
431 measurable, bounded and supported on $\mathcal{C}_{a^*}$, so it is integrable on $\mathcal{A}$.

432 • Dominated convergence ensures that, for each $k \in [\![1, m]\!]$, partial derivative $g_k(\theta) =$
433 $\partial_{\theta_k} \int_{\mathcal{A}} \nabla_\theta \pi_{\theta,\sigma}(a|s) P(s'|s,a) \mathrm{d}a$ is continuous: let $\theta_n \downarrow \theta$, then

$$g_k(\theta_n) = \partial_{\theta_k} \int_{\mathcal{A}} \nabla_\theta \pi_{\theta_n,\sigma}(a|s) P(s'|s,a) \mathrm{d}a$$

$$= \partial_{\theta_k} \mu_{\theta_n}(s) \int_{\mathcal{C}_{a^*}} \nu_\sigma(a^*, a - \mu_{\theta_n}(s) + a^*) \nabla_a P(s'|s,a) \mathrm{d}a$$

$$\xrightarrow[n \to \infty]{} \partial_{\theta_k} \mu_\theta(s) \int_{\mathcal{C}_{a^*}} \nu_\sigma(a^*, a - \mu_\theta(s) + a^*) \nabla_a P(s'|s,a) \mathrm{d}a = g_k(\theta)$$

434 with the dominating function $a \mapsto \sup_{a \in \mathcal{C}_{a^*}} |\nu_\sigma(a^*, a)| \sup_{a \in \mathcal{A}} \|\nabla_a P(s'|s,a)\| \, \mathbf{1}_{a \in \mathcal{C}_{a^*}}$.

435 Thus $\sigma \mapsto \nabla_\theta P_{s,s'}^{\pi_{\theta,\sigma}}$ is defined for $\sigma \in [0, r]$ and is continuous at 0, with $\nabla_\theta P_{s,s'}^{\pi_{\theta,0}} =$
436 $\nabla_\theta \mu_\theta(s) \, \nabla_a P(s'|s,a)|_{a=\mu_\theta(s)}$. Indeed, let $(\sigma_n)_{n \in \mathbb{N}} \in [0, r]^{+\mathbb{N}}$, $\sigma_n \downarrow 0$, then, applying the first
437 condition of Conditions 1 with $f : a \mapsto \nabla_a P(s'|s,a)$ belonging to $\mathcal{F}$, we get

$$\left\| \nabla_\theta P_{s,s'}^{\pi_{\theta,\sigma_n}} - \nabla_\theta P_{s,s'}^{\mu_\theta} \right\|$$

$$= \|\nabla_\theta \mu_\theta(s)\|_{\mathrm{op}} \left\| \int_{\mathcal{C}_{\mu_\theta(s)}} \nu_{\sigma_n}(\mu_\theta(s), a) \nabla_a P(s'|s,a) \mathrm{d}a - \nabla_a P(s'|s,a)|_{a=\mu_\theta(s)} \right\| \xrightarrow[n \to \infty]{} 0.$$

438 Since $d^{\pi_{\theta,\sigma}} = \frac{1}{\det(A^{\pi_{\theta,\sigma}})} \mathrm{adj}\left(A^{\pi_{\theta,\sigma}}\right)^\top a$ with $|\det\left(A^{\pi_{\theta,\sigma}}\right)| > 0$ for all $\sigma \in [0, r]$ and since entries
439 of $\mathrm{adj}\left(A^{\pi_{\theta,\sigma}}\right)$ and $\det\left(A^{\pi_{\theta,\sigma}}\right)$ are polynomial functions of the entries of $P^{\pi_{\theta,\sigma}}$, it follows that

440    $\sigma \mapsto \nabla_\theta d^{\pi_{\theta,\sigma}}$ is properly defined on $[0,r]$ and is continuous at 0, which concludes the proof of
441    Lemma 2. $\hfill\square$

442    We now proceed to prove Theorem 8.

443    Let $\theta \in \Theta$, $\pi_\theta$ as in Theorem 3, and $r > 0$ such that $\sigma \mapsto d^{\pi_{\theta,\sigma}}$, $\sigma \mapsto \nabla_\theta d^{\pi_{\theta,\sigma}}$ are well defined on
444    $[0,r]$ and are continuous at 0. Then, the following two functions

$$\sigma \mapsto J_{\pi_{\theta,\sigma}}(\pi_{\theta,\sigma}) = \sum_{s\in\mathcal{S}} d^{\pi_{\theta,\sigma}}(s) \int_{\mathcal{A}} \pi_{\theta,\sigma}(a|s)\bar{R}(s,a)\mathrm{d}a,$$

$$\sigma \mapsto J_{\pi_{\theta,\sigma}}(\mu_\theta) = \sum_{s\in\mathcal{S}} d^{\pi_{\theta,\sigma}}(s)\bar{R}(s,\mu_\theta(s)),$$

445    are properly defined on $[0,r]$ (with $J_{\pi_{\theta,0}}(\pi_{\theta,0}) = J_{\pi_{\theta,0}}(\mu_\theta) = J_{\mu_\theta}(\mu_\theta)$). Let $s \in \mathcal{S}$, by taking
446    similar arguments as in the proof of Lemma 2, we have

$$\nabla_\theta \int_{\mathcal{A}} \pi_{\theta,\sigma}(a|s)\bar{R}(s,a)\mathrm{d}a = \int_{\mathcal{A}} \nabla_\theta \pi_{\theta,\sigma}(a,s)\bar{R}(s,a)\mathrm{d}a,$$

$$= \nabla_\theta \mu_\theta(s) \int_{\mathcal{C}_{\mu_\theta(s)}} \nu_\sigma(\mu_\theta(s),a)\nabla_a \bar{R}(s,a)\mathrm{d}a.$$

447    Thus, $\sigma \mapsto \nabla_\theta J_{\pi_{\theta,\sigma}}(\pi_{\theta,\sigma})$ is properly defined on $[0,r]$ and

$$\nabla_\theta J_{\pi_{\theta,\sigma}}(\pi_{\theta,\sigma}) = \sum_{s\in\mathcal{S}} \nabla_\theta d^{\pi_{\theta,\sigma}}(s) \int_{\mathcal{A}} \pi_{\theta,\sigma}(a|s)\bar{R}(s,a)\mathrm{d}a$$

$$+ \sum_{s\in\mathcal{S}} d^{\pi_{\theta,\sigma}}(s) \nabla_\theta \int_{\mathcal{A}} \pi_{\theta,\sigma}(a|s)\bar{R}(s,a)\mathrm{d}a$$

$$= \sum_{s\in\mathcal{S}} \nabla_\theta d^{\pi_{\theta,\sigma}}(s) \int_{\mathcal{A}} \nu_\sigma(\mu_\theta(s),a)\bar{R}(s,a)\mathrm{d}a$$

$$+ \sum_{s\in\mathcal{S}} d^{\pi_{\theta,\sigma}}(s) \nabla_\theta \mu_\theta(s) \int_{\mathcal{C}_{\mu_\theta(s)}} \nu_\sigma(\mu_\theta(s),a)\nabla_a \bar{R}(s,a)\mathrm{d}a.$$

448    Similarly, $\sigma \mapsto \nabla_\theta J_{\pi_{\theta,\sigma}}(\mu_\theta)$ is properly defined on $[0,r]$ and

$$\nabla_\theta J_{\pi_{\theta,\sigma}}(\mu_\theta) = \sum_{s\in\mathcal{S}} \nabla_\theta d^{\pi_{\theta,\sigma}}(s)\bar{R}(s,\mu_\theta(s)) + \sum_{s\in\mathcal{S}} d^{\pi_{\theta,\sigma}}(s) \nabla_\theta \mu_\theta(s)\, \nabla_a \bar{R}(s,a)\big|_{a=\mu_\theta(s)}$$

449    To prove continuity at 0 of both $\sigma \mapsto \nabla_\theta J_{\pi_{\theta,\sigma}}(\pi_{\theta,\sigma})$ and $\sigma \mapsto \nabla_\theta J_{\pi_{\theta,\sigma}}(\mu_\theta)$ (with $\nabla_\theta J_{\pi_{\theta,0}}(\pi_{\theta,0}) =$
450    $\nabla_\theta J_{\pi_{\theta,0}}(\mu_\theta) = \nabla_\theta J_{\mu_\theta}(\mu_\theta)$), let $(\sigma_n)_{n\geq 0} \downarrow 0$:

$$\left\| \nabla_\theta J_{\pi_{\theta,\sigma_n}}(\pi_{\theta,\sigma_n}) - \nabla_\theta J_{\pi_{\theta,0}}(\pi_{\theta,0}) \right\|$$
$$\leq \left\| \nabla_\theta J_{\pi_{\theta,\sigma_n}}(\pi_{\theta,\sigma_n}) - \nabla_\theta J_{\pi_{\theta,\sigma_n}}(\mu_\theta) \right\| + \left\| \nabla_\theta J_{\pi_{\theta,\sigma_n}}(\mu_\theta) - \nabla_\theta J_{\mu_\theta}(\mu_\theta) \right\|. \tag{19}$$

451    For the first term of the r.h.s we have

$$\left\| \nabla_\theta J_{\pi_{\theta,\sigma_n}}(\pi_{\theta,\sigma_n}) - \nabla_\theta J_{\pi_{\theta,\sigma_n}}(\mu_\theta) \right\|$$
$$\leq \sum_{s\in\mathcal{S}} \left\| \nabla_\theta d^{\pi_{\theta,\sigma_n}}(s) \right\| \left| \int_{\mathcal{A}} \nu_{\sigma_n}(\mu_\theta(s),a)\bar{R}(s,a)\mathrm{d}a - \bar{R}(s,\mu_\theta(s)) \right|$$
$$+ \sum_{s\in\mathcal{S}} d^{\pi_{\theta,\sigma_n}}(s) \left\| \nabla_\theta \mu_\theta(s) \right\|_{\mathrm{op}} \left\| \int_{\mathcal{A}} \nu_{\sigma_n}(\mu_\theta(s),a)\nabla_a \bar{R}(s,a)\mathrm{d}a - \nabla_a \bar{R}(s,a)\big|_{a=\mu_\theta(s)} \right\|.$$

452 Applying the first assumption in Condition 1 with $f : a \mapsto \bar{R}(s, a)$ and $f : a \mapsto \nabla_a \bar{R}(s, a)$ belonging
453 to $\mathcal{F}$ we have, for each $s \in \mathcal{S}$:

$$\left| \int_{\mathcal{A}} \nu_{\sigma_n}(\mu_\theta(s), a) \bar{R}(s, a) \mathrm{d}a - \bar{R}(s, \mu_\theta(s)) \right| \xrightarrow[n \to \infty]{} 0 \quad \text{and}$$

$$\left\| \int_{\mathcal{A}} \nu_{\sigma_n}(\mu_\theta(s), a) \nabla_a \bar{R}(s, a) \mathrm{d}a - \nabla_a \bar{R}(s, a) \big|_{a=\mu_\theta(s)} \right\| \xrightarrow[n \to \infty]{} 0.$$

454 Moreover, for each $s \in \mathcal{S}$, $d^{\pi_\theta, \sigma_n}(s) \xrightarrow[n \to \infty]{} d^{\mu_\theta}(s)$ and $\nabla_\theta d^{\pi_\theta, \sigma_n}(s) \xrightarrow[n \to \infty]{} \nabla_\theta d^{\mu_\theta}(s)$ (by Lemma 2),
455 and $\|\nabla_\theta \mu_\theta(s)\|_{\mathrm{op}} < \infty$, so

$$\left\| \nabla_\theta J_{\pi_{\theta, \sigma_n}}(\pi_{\theta, \sigma_n}) - \nabla_\theta J_{\pi_{\theta, \sigma_n}}(\mu_\theta) \right\| \xrightarrow[n \to \infty]{} 0.$$

456 For the second term of the r.h.s of (19), we have

$$\left\| \nabla_\theta J_{\pi_{\theta, \sigma_n}}(\mu_\theta) - \nabla_\theta J_{\mu_\theta}(\mu_\theta) \right\| \leq \sum_{s \in \mathcal{S}} \left\| \nabla_\theta d^{\pi_\theta, \sigma_n}(s) - \nabla_\theta d^{\mu_\theta}(s) \right\| \left| \bar{R}(s, \mu_\theta(s)) \right|$$

$$+ \sum_{s \in \mathcal{S}} \left| d^{\pi_\theta, \sigma_n}(s) - d^{\mu_\theta}(s) \right| \left\| \nabla_\theta \mu_\theta(s) \right\|_{\mathrm{op}} \left\| \nabla_a \bar{R}(s, a) \big|_{a=\mu_\theta(s)} \right\|.$$

457 Continuity at 0 of $\sigma \mapsto d^{\pi_\theta, \sigma}(s)$ and $\sigma \mapsto \nabla_\theta d^{\pi_\theta, \sigma}(s)$ for each $s \in \mathcal{S}$, boundedness of $\bar{R}(s, \cdot)$,
458 $\nabla_a \bar{R}(s, \cdot)$ and $\nabla_\theta(s) \mu_\theta(s)$ implies that

$$\left\| \nabla_\theta J_{\pi_{\theta, \sigma_n}}(\mu_\theta) - \nabla_\theta J_{\mu_\theta}(\mu_\theta) \right\| \xrightarrow[n \to \infty]{} 0.$$

459 Hence,

$$\left\| \nabla_\theta J_{\pi_{\theta, \sigma_n}}(\pi_{\theta, \sigma_n}) - \nabla_\theta J_{\pi_{\theta, 0}}(\pi_{\theta, 0}) \right\| \xrightarrow[n \to \infty]{} 0.$$

460 So, $\sigma \mapsto \nabla_\theta J_{\pi_{\theta, \sigma}}(\pi_{\theta, \sigma})$ and $\nabla_\theta J_{\pi_{\theta, \sigma}}(\mu_\theta)$ are continuous at 0:

$$\lim_{\sigma \downarrow 0} \nabla_\theta J_{\pi_{\theta, \sigma}}(\pi_{\theta, \sigma}) = \lim_{\sigma \downarrow 0} \nabla_\theta J_{\pi_{\theta, \sigma}}(\mu_\theta) = \nabla_\theta J_{\mu_\theta}(\mu_\theta).$$

## Proof of Theorem 4

462 We will use the two-time-scale stochastic approximation analysis . We let the policy parameter $\theta_t$
463 fixed as $\theta_t \equiv \theta$ when analysing the convergence of the critic step. Thus we can show the convergence
464 of $\omega_t$ towards an $\omega_\theta$ depending on $\theta$, which will then be used to prove the convergence for the slow
465 time-scale.

466 **Lemma 3.** *Under Assumptions $3 - 5$, the sequence $\omega_t^i$ generated from (2) is bounded a.s., i.e.,*
467 $\sup_t \|\omega_t^i\| < \infty$ *a.s., for any $i \in \mathcal{N}$.*

468 The proof follows the same steps as that of Lemma B.1 in the PMLR version of Zhang et al. [2018].

469 **Lemma 4.** *Under Assumption 5, the sequence $\{\hat{J}_t^i\}$ generated as in 2 is bounded a.s, i.e., $\sup_t |\hat{J}_t^i| <$*
470 $\infty$ *a.s., for any $i \in \mathcal{N}$.*

471 The proof follows the same steps as that of Lemma B.2 in the PMLR version of Zhang et al. [2018].

472 The desired result holds since **Step 1** and **Step 2** of the proof of Theorem 4.6 in Zhang et al. [2018]
473 can both be repeated in the setting of deterministic policies.

## Proof of Theorem 5

475 Let $\mathcal{F}_{t,2} = \sigma(\theta_\tau, s_\tau, \tau \leq t)$ a filtration. In addition, we define

$$H(\theta, s, \omega) = \nabla_\theta \mu_\theta(s) \cdot \nabla_a Q_\omega(s, a) \big|_{a=\mu_\theta(s)},$$
$$H(\theta, s) = H(\theta, s, \omega_\theta),$$
$$h(\theta) = \mathbb{E}_{s \sim d^\theta} \left[ H(\theta, s) \right].$$

476  Then, for each $\theta \in \Theta$, we can introduce $\nu_\theta : \mathcal{S} \to \mathbb{R}^n$ the solution to the Poisson equation:

$$\left(I - P^\theta\right)\nu_\theta(\cdot) = H(\theta, \cdot) - h(\theta)$$

477  that is given by $\nu_\theta(s) = \sum_{k \geq 0} \mathbb{E}_{s_{k+1} \sim P^\theta(\cdot|s_k)}\left[H(\theta, s_k) - h(\theta)|s_0 = s\right]$ which is properly defined
478  (similar to the differential value function $V$).

479  With projection, actor update (5) becomes

$$
\begin{aligned}
\theta_{t+1} &= \Gamma\left[\theta_t + \beta_{\theta,t} H(\theta_t, s_t, \omega_t)\right] &&(20)\\
&= \Gamma\left[\theta_t + \beta_{\theta,t} h(\theta_t) - \beta_{\theta,t}\left(h(\theta_t) - H(\theta_t, s_t)\right) - \beta_{\theta,t}\left(H(\theta_t, s_t) - H(\theta_t, s_t, \omega_t)\right)\right]\\
&= \Gamma\left[\theta_t + \beta_{\theta,t} h(\theta_t) + \beta_{\theta,t}\left((I - P^{\theta_t})\nu_{\theta_t}(s_t)\right) + \beta_{\theta,t} A_t^1\right]\\
&= \Gamma\left[\theta_t + \beta_{\theta,t} h(\theta_t) + \beta_{\theta,t}\left(\nu_{\theta_t}(s_t) - \nu_{\theta_t}(s_{t+1})\right) + \beta_{\theta,t}\left(\nu_{\theta_t}(s_{t+1}) - P^{\theta_t}\nu_{\theta_t}(s_t)\right) + \beta_{\theta,t} A_t^1\right]\\
&= \Gamma\left[\theta_t + \beta_{\theta,t}\left(h(\theta_t) + A_t^1 + A_t^2 + A_t^3\right)\right]
\end{aligned}
$$

480  where

$$
\begin{aligned}
A_t^1 &= H(\theta_t, s_t, \omega_t) - H(\theta_t, s_t),\\
A_t^2 &= \nu_{\theta_t}(s_t) - \nu_{\theta_t}(s_{t+1}),\\
A_t^3 &= \nu_{\theta_t}(s_{t+1}) - P^{\theta_t}\nu_{\theta_t}(s_t).
\end{aligned}
$$

481  For $r < t$ we have

$$
\begin{aligned}
\sum_{k=r}^{t-1}\beta_{\theta,k}A_k^2 &= \sum_{k=r}^{t-1}\beta_{\theta,k}\left(\nu_{\theta_k}(s_k) - \nu_{\theta_k}(s_{k+1})\right)\\
&= \sum_{k=r}^{t-1}\beta_{\theta,k}\left(\nu_{\theta_k}(s_k) - \nu_{\theta_{k+1}}(s_{k+1})\right) + \sum_{k=r}^{t-1}\beta_{\theta,k}\left(\nu_{\theta_{k+1}}(s_{k+1}) - \nu_{\theta_k}(s_{k+1})\right)\\
&= \sum_{k=r}^{t-1}\left(\beta_{\theta,k+1} - \beta_{\theta,k}\right)\nu_{\theta_{k+1}}(s_{k+1}) + \beta_{\theta_r}\nu_{\theta_r}(s_r) - \beta_{\theta_t}\nu_{\theta_t}(s_t) + \sum_{k=r}^{t-1}\epsilon_k^{(2)}\\
&= \sum_{k=r}^{t-1}\epsilon_k^{(1)} + \sum_{k=r}^{t-1}\epsilon_k^{(2)} + \eta_{r,t}
\end{aligned}
$$

482  where

$$
\begin{aligned}
\epsilon_k^{(1)} &= \left(\beta_{\theta,k+1} - \beta_{\theta,k}\right)\nu_{\theta_{k+1}}(s_{k+1}),\\
\epsilon_k^{(2)} &= \beta_{\theta,k}\left(\nu_{\theta_{k+1}}(s_{k+1}) - \nu_{\theta_k}(s_{k+1})\right),\\
\eta_{r,t} &= \beta_{\theta_r}\nu_{\theta_r}(s_r) - \beta_{\theta_t}\nu_{\theta_t}(s_t).
\end{aligned}
$$

483  **Lemma 5.** $\sum_{k=0}^{t-1}\beta_{\theta,k}A_k^2$ *converges a.s. for* $t \to \infty$

484  *Proof of Lemma 5.* Since $\nu_\theta(s)$ is uniformly bounded for $\theta \in \Theta, s \in \mathcal{S}$, we have for some $K > 0$

$$\sum_{k=0}^{t-1}\left\|\epsilon_k^{(1)}\right\| \leq K\sum_{k=0}^{t-1}\left|\beta_{\theta,k+1} - \beta_{\theta,k}\right|$$

485  which converges given Assumption 5.

486  Moreover, since $\mu_\theta(s)$ is twice continuously differentiable, $\theta \mapsto \nu_\theta(s)$ is Lipschitz for each $s$, and so
487  we have

$$
\begin{aligned}
\sum_{k=0}^{t-1}\left\|\epsilon_k^{(2)}\right\| &\leq \sum_{k=0}^{t-1}\beta_{\theta,k}\left\|\nu_{\theta_k}(s_{k+1}) - \nu_{\theta_{k+1}}(s_{k+1})\right\|\\
&\leq K^2\sum_{k=0}^{t-1}\beta_{\theta,k}\left\|\theta_k - \theta_{k+1}\right\|\\
&\leq K^3\sum_{k=0}^{t-1}\beta_{\theta,k}^2.
\end{aligned}
$$

488 Finally, $\lim_{t \to \infty} \|\eta_{0,t}\| = \beta_{\theta,0} \|\nu_{\theta_0}(s_0)\| < \infty$ a.s.

489 Thus, $\sum_{k=0}^{t-1} \|\beta_{\theta,k} A_k^2\| \le \sum_{k=0}^{t-1} \left\|\epsilon_k^{(1)}\right\| + \sum_{k=0}^{t-1} \left\|\epsilon_k^{(2)}\right\| + \|\eta_{0,t}\|$ converges a.s. $\qquad \square$

490 **Lemma 6.** $\sum_{k=0}^{t-1} \beta_{\theta,k} A_k^3$ *converges a.s. for* $t \to \infty$.

491 *Proof of Lemma 6.* We set

$$Z_t = \sum_{k=0}^{t-1} \beta_{\theta,k} A_k^3 = \sum_{k=0}^{t-1} \beta_{\theta,k} \left(\nu_{\theta_k}(s_{k+1}) - P^{\theta_k} \nu_{\theta_k}(s_k)\right).$$

492 Since $Z_t$ is $\mathcal{F}_t$-adapted and $\mathbb{E}\left[\nu_{\theta_t}(s_{t+1})|\mathcal{F}_t\right] = P^{\theta_t} \nu_{\theta_t}(s_t)$, $Z_t$ is a martingale. The remaining of the
493 proof is now similar to the proof of Lemma 2 on page 224 of Benveniste et al. [1990]. $\qquad \square$

Let $g^i(\theta_t) = \mathbb{E}_{s_t \sim d^{\theta_t}}\left[\psi_t^i \cdot \xi_{t,\theta_t}^i | \mathcal{F}_{t,2}\right]$ and $g(\theta) = \left[g^1(\theta), \ldots, g^N(\theta)\right]$. We have

$$g^i(\theta_t) = \sum_{s_t \in \mathcal{S}} d^{\theta_t}(s_t) \cdot \psi_t^i \cdot \xi_{t,\theta_t}^i.$$

494 Given (10), $\theta \mapsto \omega_\theta$ is continuously differentiable and $\theta \mapsto \nabla_\theta \omega_\theta$ is bounded so $\theta \mapsto \omega_\theta$ is
495 Lipschitz-continuous. Thus $\theta \mapsto \xi_{t,\theta}^i$ is Lipschitz-continuous for each $s_t \in \mathcal{S}$. Due to our regularity
496 assumptions, $\theta \mapsto \psi_{t,\theta_t}^i$ is also continuous for each $i \in \mathcal{N}, s_t \in \mathcal{S}$. Moreover, $\theta \mapsto d^\theta(s)$ is also
497 Lipschitz continuous for each $s \in \mathcal{S}$. Hence, $\theta \mapsto g(\theta)$ is Lipschitz-continuous in $\theta$ and the ODE
498 (12) is well-posed. This holds even when using compatible features.

499 By critic faster convergence, we have $\lim_{t \to \infty} \|\xi_t^i - \xi_{t,\theta_t}^i\| = 0$ so $\lim_{t \to \infty} A_t^1 = 0$.

500 Hence, by Kushner-Clark lemma Kushner and Clark [1978] (pp 191-196) we have that the update in
501 (20) converges a.s. to the set of asymptotically stable equilibria of the ODE (12).

## Proof of Theorem 6

503 We use the two-time scale technique: since critic updates at a faster rate than the actor, we let the
504 policy parameter $\theta_t$ to be fixed as $\theta$ when analysing the convergence of the critic update.

505 **Lemma 7.** *Under Assumptions 4, 1 and 6, for any* $i \in \mathcal{N}$, *sequence* $\{\lambda_t^i\}$ *generated from (7) is*
506 *bounded almost surely.*

507 To prove this lemma we verify the conditions for **Theorem A.2** of Zhang et al. [2018] to hold.
508 We use $\{\mathcal{F}_{t,1}\}$ to denote the filtration with $\mathcal{F}_{t,1} = \sigma(s_\tau, C_{\tau-1}, a_{\tau-1}, r_\tau, \lambda_\tau, \tau \le t)$. With $\lambda_t =$
509 $\left[(\lambda_t^1)^\top, \ldots, (\lambda_t^N)^\top\right]^\top$, critic step (7) has the form:

$$\lambda_{t+1} = (C_t \otimes I)(\lambda_t + \beta_{\lambda,t} \cdot y_{t+1}) \tag{21}$$

510 with $y_{t+1} = \left(\delta_t^1 w(s_t, a_t)^\top, \ldots, \delta_t^N w(s_t, a_t)^\top\right)^\top \in \mathbb{R}^{KN}$, $\otimes$ denotes Kronecker product and $I$ is
511 the identity matrix. Using the same notation as in **Assumption A.1** from Zhang et al. [2018], we
512 have:

$$h^i(\lambda_t^i, s_t) = \mathbb{E}_{a \sim \pi}\left[\delta_t^i w(s_t, a)^\top | \mathcal{F}_{t,1}\right] = \int_{\mathcal{A}} \pi(a|s_t)(R^i(s_t, a) - w(s_t, a) \cdot \lambda_t^i)w(s_t, a)^\top \mathrm{d}a,$$

$$M_{t+1}^i = \delta_t^i w(s_t, a_t)^\top - \mathbb{E}_{a \sim \pi}\left[\delta_t^i w(s_t, a)^\top | \mathcal{F}_{t,1}\right],$$

$$\bar{h}^i(\lambda_t) = A_{\pi,\theta}^i \cdot d_\pi^s - B_{\pi,\theta} \cdot \lambda_t, \qquad \text{where } A_{\pi,\theta}^i = \left[\int_{\mathcal{A}} \pi(a|s)R^i(s,a)w(s,a)^\top \mathrm{d}a, s \in \mathcal{S}\right].$$

513 Since feature vectors are uniformly bounded for any $s \in \mathcal{S}$ and $a \in \mathcal{A}$, $h^i$ is Lipschitz continuous in
514 its first argument. Since, for $i \in \mathcal{N}$, the $r^i$ are also uniformly bounded, $\mathbb{E}\left[\|M_{t+1}\|^2 | \mathcal{F}_{t,1}\right] \le K \cdot (1 +$
515 $\|\lambda_t\|^2)$ for some $K > 0$. Furthermore, finiteness of $|\mathcal{S}|$ ensures that, a.s., $\|\bar{h}(\lambda_t) - h(\lambda_t, s_t)\|^2 \le$
516 $K' \cdot (1 + \|\lambda_t\|^2)$. Finally, $h_\infty(y)$ exists and has the form

$$h_\infty(y) = -B_{\pi,\theta} \cdot y.$$

517 From Assumption 1, we have that $-B_{\pi,\theta}$ is a Hurwitcz matrix, thus the origin is a globally asymptot-
518 ically stable attractor of the ODE $\dot{y} = h_\infty(y)$. Hence **Theorem A.2** of Zhang et al. [2018] applies,
519 which concludes the proof of Lemma 7.

520  We introduce the following operators as in Zhang et al. [2018]:

521  • $\langle \cdot \rangle : \mathbb{R}^{KN} \to \mathbb{R}^K$

$$\langle \lambda \rangle = \frac{1}{N}(\mathbf{1}^\top \otimes I)\lambda = \frac{1}{N}\sum_{i \in \mathcal{N}} \lambda^i.$$

522  • $\mathcal{J} = \left(\frac{1}{N}\mathbf{1}\mathbf{1}^\top \otimes I\right) : \mathbb{R}^{KN} \to \mathbb{R}^{KN}$ such that $\mathcal{J}\lambda = \mathbf{1} \otimes \langle \lambda \rangle$.

523  • $\mathcal{J}_\perp = I - \mathcal{J} : \mathbb{R}^{KN} \to \mathbb{R}^{KN}$ and we note $\lambda_\perp = \mathcal{J}_\perp \lambda = \lambda - \mathbf{1} \otimes \langle \lambda \rangle$.

524  We then proceed in two steps as in Zhang et al. [2018], firstly by showing the convergence a.s. of the
525  disagreement vector sequence $\{\lambda_{\perp,t}\}$ to zero, secondly showing that the consensus vector sequence
526  $\{\langle \lambda_t \rangle\}$ converges to the equilibrium such that $\langle \lambda_t \rangle$ is solution to (13).

527  **Lemma 8.** *Under Assumptions 4, 1 and 6, for any $M > 0$, we have*

$$\sup_t \mathbb{E}\Big[\|\beta_{\lambda,t}^{-1}\lambda_{\perp,t}\|^2 \mathbb{1}_{\{\sup_t \|\lambda_t\| \leq M\}}\Big] < \infty.$$

528  Since dynamic of $\{\lambda_t\}$ described by (21) is similar to (5.2) in Zhang et al. [2018] we have

$$\mathbb{E}\Big[\|\beta_{\lambda,t+1}^{-1}\lambda_{\perp,t+1}\|^2|\mathcal{F}_{t,1}\Big] = \frac{\beta_{\lambda,t}^2}{\beta_{\lambda,t+1}^2}\rho\left(\|\beta_{\lambda,t}^{-1}\lambda_{\perp,t}\|^2 + 2\cdot\|\beta_{\lambda,t}^{-1}\lambda_{\perp,t}\|\cdot\mathbb{E}(\|y_{t+1}\|^2|\mathcal{F}_{t,1})^{\frac{1}{2}} + \mathbb{E}(\|y_{t+1}\|^2|\mathcal{F}_{t,1})\right)$$

$$(22)$$

529  where $\rho$ represents the spectral norm of $\mathbb{E}\big[C_t^\top \cdot (I - \mathbf{1}\mathbf{1}^\top/N)\cdot C_t\big]$, with $\rho \in [0,1)$ by Assumption
530  4. Since $y_{t+1}^i = \delta_t^i \cdot w(s_t, a_t)^\top$ we have

$$\mathbb{E}\Big[\|y_{t+1}\|^2|\mathcal{F}_{t,1}\Big] = \mathbb{E}\Big[\sum_{i \in \mathcal{N}}\|(r^i(s_t,a_t) - w(s_t,a_t)\lambda_t^i)\cdot w(s_t,a_t)^\top\|^2|\mathcal{F}_{t,1}\Big]$$

$$\leq 2\cdot\mathbb{E}\Big[\sum_{i \in \mathcal{N}}\|r^i(s_t,a_t)w(s_t,a_t)^\top\|^2 + \|w(s_t,a_t)^\top\|^4\cdot\|\lambda_t^i\|^2|\mathcal{F}_{t,1}\Big].$$

531  By uniform boundedness of $r(s,\cdot)$ and $w(s,\cdot)$ (Assumptions 1) and finiteness of $\mathcal{S}$, there exists
532  $K_1 > 0$ such that

$$\mathbb{E}\Big[\|y_{t+1}\|^2|\mathcal{F}_{t,1}\Big] \leq K_1(1 + \|\lambda_t\|^2).$$

533  Thus, for any $M > 0$ there exists $K_2 > 0$ such that, on the set $\{\sup_{\tau \leq t}\|\lambda_\tau\| < M\}$,

$$\mathbb{E}\Big[\|y_{t+1}\|^2 \mathbb{1}_{\{\sup_{\tau \leq t}\|\lambda_\tau\| < M\}}|\mathcal{F}_{t,1}\Big] \leq K_2. \tag{23}$$

534  We let $v_t = \|\beta_{\lambda,t}^{-1}\lambda_{\perp,t}\|^2\mathbb{1}_{\{\sup_{\tau \leq t}\|\lambda_\tau\| < M\}}$.  Taking expectation over (22), noting that
535  $\mathbb{1}_{\{\sup_{\tau \leq t+1}\|\lambda_\tau\| < M\}} \leq \mathbb{1}_{\{\sup_{\tau \leq t}\|\lambda_\tau\| < M\}}$ we get

$$\mathbb{E}(v_{t+1}) \leq \frac{\beta_{\lambda,t}^2}{\beta_{\lambda,t+1}^2}\rho\left(\mathbb{E}(v_t) + 2\sqrt{\mathbb{E}(v_t)}\cdot\sqrt{K_2} + K_2\right)$$

536  which is the same expression as (5.10) in Zhang et al. [2018]. So similar conclusions to the ones of
537  **Step 1** of Zhang et al. [2018] holds:

$$\sup_t \mathbb{E}\Big[\|\beta_{\lambda,t}^{-1}\lambda_{\perp,t}\|^2\mathbb{1}_{\{\sup_t\|\lambda_t\| \leq M\}}\Big] < \infty \tag{24}$$

$$\text{and} \qquad \lim_t \lambda_{\perp,t} = 0 \text{ a.s.} \tag{25}$$

538  We now show convergence of the consensus vector $\mathbf{1} \otimes \langle \lambda_t \rangle$. Based on (21) we have

$$\langle \lambda_{t+1} \rangle = \langle (C_t \otimes I)(\mathbf{1} \otimes \langle \lambda_t \rangle + \lambda_{\perp,t} + \beta_{\lambda,t}y_{t+1})\rangle$$

$$= \langle \lambda_t \rangle + \langle \lambda_{\perp,t} \rangle + \beta_{\lambda,t}\langle (C_t \otimes I)(y_{t+1} + \beta_{\lambda,t}^{-1}\lambda_{\perp,t})\rangle$$

$$= \langle \lambda_t \rangle + \beta_{\lambda,t}(h(\lambda_t, s_t) + M_{t+1})$$

539 where $h(\lambda_t, s_t) = \mathbb{E}_{a_t \sim \pi}\big[\langle y_{t+1}\rangle | \mathcal{F}_t\big]$ and $M_{t+1} = \langle (C_t \otimes I)(y_{t+1} + \beta_{\lambda,t}^{-1}\lambda_{\perp,t})\rangle - \mathbb{E}_{a_t \sim \pi}\big[\langle y_{t+1}\rangle | \mathcal{F}_t\big]$.

540 Since $\langle \delta_t \rangle = \bar{r}(s_t, a_t) - w(s_t, a_t)\langle \lambda_t \rangle$, we have

$$h(\lambda_t, s_t) = \mathbb{E}_{a_t \sim \pi}(\bar{r}(s_t, a_t) w(s_t, a_t)^\top | \mathcal{F}_t) + \mathbb{E}_{a_t \sim \pi}(w(s_t, a_t)\langle \lambda_t \rangle \cdot w(s_t, a_t)^\top | \mathcal{F}_{t,1})$$

541 so $h$ is Lipschitz-continuous in its first argument. Moreover, since $\langle \lambda_{\perp,t} \rangle = 0$ and $\mathbf{1}^\top \mathbb{E}(C_t | \mathcal{F}_{t,1}) =$
542 $\mathbf{1}^\top$ a.s.:

$$
\begin{aligned}
\mathbb{E}_{a_t \sim \pi}\big[\langle (C_t \otimes I)(y_{t+1} + \beta_{\lambda,t}^{-1}\lambda_{\perp,t})\rangle | \mathcal{F}_{t,1}\big] &= \mathbb{E}_{a_t \sim \pi}\Big[\frac{1}{N}(\mathbf{1}^\top \otimes I)(C_t \otimes I)(y_{t+1} + \beta_{\lambda,t}^{-1}\lambda_{\perp,t}) | \mathcal{F}_{t,1}\Big] \\
&= \frac{1}{N}(\mathbf{1}^\top \otimes I)(\mathbb{E}(C_t | \mathcal{F}_{t,1}) \otimes I)\mathbb{E}_{a_t \sim \pi}\big[y_{t+1} + \beta_{\lambda,t}^{-1}\lambda_{\perp,t} | \mathcal{F}_{t,1}\big] \\
&= \frac{1}{N}(\mathbf{1}^\top \mathbb{E}(C_t | \mathcal{F}_{t,1}) \otimes I)\mathbb{E}_{a_t \sim \pi}\big[y_{t+1} + \beta_{\lambda,t}^{-1}\lambda_{\perp,t} | \mathcal{F}_{t,1}\big] \\
&= \mathbb{E}_{a_t \sim \pi}\big[\langle y_{t+1}\rangle | \mathcal{F}_{t,1}\big] \text{ a.s.}
\end{aligned}
$$

543 So $\{M_t\}$ is a martingale difference sequence. Additionally we have

$$\mathbb{E}\big[\|M_{t+1}\|^2 | \mathcal{F}_{t,1}\big] \leq 2 \cdot \mathbb{E}\big[\|y_{t+1} + \beta_{\lambda,t}^{-1}\lambda_{\perp,t}\|_{G_t}^2 | \mathcal{F}_{t,1}\big] + 2 \cdot \|\mathbb{E}\big[\langle y_{t+1}\rangle | \mathcal{F}_{t,1}\big]\|^2$$

544 with $G_t = N^{-2} \cdot C_t^\top \mathbf{1}\mathbf{1}^\top C_t \otimes I$ whose spectral norm is bounded for $C_t$ is stochastic. From (23) and
545 (24) we have that, for any $M > 0$, over the set $\{\sup_t \|\lambda_t\| \leq M\}$, there exists $K_3, K_4 < \infty$ such that

$$\mathbb{E}\big[\|y_{t+1} + \beta_{\lambda,t}^{-1}\lambda_{\perp,t}\|_{G_t}^2 | \mathcal{F}_{t,1}\big] \mathbb{1}_{\{\sup_t \|\lambda_t\| \leq M\}} \leq K_3 \cdot \mathbb{E}\big[\|y_{t+1}\|^2 + \|\beta_{\lambda,t}^{-1}\lambda_{\perp,t}\|^2 | \mathcal{F}_{t,1}\big] \mathbb{1}_{\{\sup_t \|\lambda_t\| \leq M\}} \leq K_4.$$

546 Besides, since $r_{t+1}^i$ and $w$ are uniformly bounded, there exists $K_5 < \infty$ such that
547 $\|\mathbb{E}\big[\langle y_{t+1}\rangle | \mathcal{F}_{t,1}\big]\|^2 \leq K_5 \cdot (1 + \|\langle \lambda_t \rangle\|^2)$. Thus, for any $M > 0$, there exists some $K_6 < \infty$
548 such that over the set $\{\sup_t \|\lambda_t\| \leq M\}$

$$\mathbb{E}\big[\|M_{t+1}\|^2 | \mathcal{F}_{t,1}\big] \leq K_6 \cdot (1 + \|\langle \lambda_t \rangle\|^2).$$

549 Hence, for any $M > 0$, assumptions (a.1) - (a.5) of B.1. from Zhang et al. [2018] are verified on the
550 set $\{\sup_t \|\lambda_t\| \leq M\}$. Finally, we consider the ODE asymptotically followed by $\langle \lambda_t \rangle$:

$$\langle \dot{\lambda}_t \rangle = -B_{\pi,\theta} \cdot \langle \lambda_t \rangle + A_{\pi,\theta} \cdot d^\pi$$

551 which has a single globally asymptotically stable equilibrium $\lambda^* \in \mathbb{R}^K$, since $B_{\pi,\theta}$ is positive
552 definite: $\lambda^* = B_{\pi,\theta}^{-1} \cdot A_{\pi,\theta} \cdot d^\pi$. By Lemma 7, $\sup_t \|\langle \lambda_t \rangle\| < \infty$ a.s., all conditions to apply **Theorem**
553 **B.2.** of Zhang et al. [2018] hold a.s., which means that $\langle \lambda_t \rangle \underset{t \to \infty}{\longrightarrow} \lambda^*$ a.s. As $\lambda_t = \mathbf{1} \otimes \langle \lambda_t \rangle + \lambda_{\perp,t}$
554 and $\lambda_{\perp,t} \underset{t \to \infty}{\longrightarrow} 0$ a.s., we have for each $i \in \mathcal{N}$, a.s.,

$$\lambda_t^i \underset{t \to \infty}{\longrightarrow} B_{\pi,\theta}^{-1} \cdot A_{\pi,\theta} \cdot d^\pi.$$

555 **Proof of Theorem 7**

556 Let $\mathcal{F}_{t,2} = \sigma(\theta_\tau, \tau \leq t)$ be the $\sigma$-field generated by $\{\theta_\tau, \tau \leq t\}$, and let

$$\zeta_{t,1}^i = \psi_t^i \cdot \xi_t^i - \mathbb{E}_{s_t \sim d^\pi}\big[\psi_t^i \cdot \xi_t^i | \mathcal{F}_{t,2}\big], \qquad \zeta_{t,2}^i = \mathbb{E}_{s_t \sim d^\pi}\big[\psi_t^i \cdot (\xi_t^i - \xi_{t,\theta_t}^i) | \mathcal{F}_{t,2}\big].$$

557 With local projection, actor update (6) becomes

$$\theta_{t+1}^i = \Gamma^i\big[\theta_t^i + \beta_{\theta,t}\mathbb{E}_{s_t \sim d^\pi}\big[\psi_t^i \cdot \xi_{t,\theta_t}^i | \mathcal{F}_{t,2}\big] + \beta_{\theta,t}\zeta_{t,1}^i + \beta_{\theta,t}\zeta_{t,2}^i\big]. \tag{26}$$

So with $h^i(\theta_t) = \mathbb{E}_{s_t \sim d^\pi}\big[\psi_t^i \cdot \xi_{t,\theta_t}^i | \mathcal{F}_{t,2}\big]$ and $h(\theta) = \big[h^1(\theta), \dots, h^N(\theta)\big]$, we have

$$h^i(\theta_t) = \sum_{s_t \in \mathcal{S}} d^\pi(s_t) \cdot \psi_t^i \cdot \xi_{t,\theta_t}^i.$$

558 Given (10), $\theta \mapsto \omega_\theta$ is continuously differentiable and $\theta \mapsto \nabla_\theta \omega_\theta$ is bounded so $\theta \mapsto \omega_\theta$ is Lipschitz-
559 continuous. Thus $\theta \mapsto \xi_{t,\theta}^i$ is Lipschitz-continuous for each $s_t \in \mathcal{S}$. Our regularity assumptions

ensure that $\theta \mapsto \psi^i_{t,\theta_t}$ is continuous for each $i \in \mathcal{N}, s_t \in \mathcal{S}$. Moreover, $\theta \mapsto d^\theta(s)$ is also Lipschitz continuous for each $s \in \mathcal{S}$. Hence, $\theta \mapsto g(\theta)$ is Lipschitz-continuous in $\theta$ and the ODE (12) is well-posed. This holds even when using compatible features.

By critic faster convergence, we have $\lim_{t\to\infty} \|\xi^i_t - \xi^i_{t,\theta_t}\| = 0$.

Let $M^i_t = \sum_{\tau=0}^{t-1} \beta_{\theta,\tau} \zeta^i_{\tau,1}$. $M^i_t$ is a martingale sequence with respect to $\mathcal{F}_{t,2}$. Since $\{\omega_t\}_t, \{\nabla_a \phi_k(s,a)\}_{s,k}$, and $\{\nabla_\theta \mu_\theta(s)\}_s$ are bounded (Lemma 3, Assumption 2), it follows that the sequence $\{\zeta^i_{t,1}\}$ is bounded. Thus, by Assumption 5, $\sum_t \mathbb{E}\left[ \left\| M^i_{t+1} - M^i_t \right\|^2 | \mathcal{F}_{t,2} \right] = \sum_t \left\| \beta_{\theta,t} \zeta^i_{t,1} \right\|^2 < \infty$ a.s. The martingale convergence theorem ensures that $\{M^i_t\}$ converges a.s. Thus, for any $\epsilon > 0$,

$$\lim_t \mathbb{P}\left( \sup_{n \geq t} \left\| \sum_{\tau=t}^n \beta_{\theta,\tau} \zeta^i_{\tau,1} \right\| \geq \epsilon \right) = 0.$$

Hence, by Kushner-Clark lemma Kushner and Clark [1978] (pp 191-196) we have that the update in (26) converges a.s. to the set of asymptotically stable equilibria of the ODE (12).

