# OpenReview forum: "Decentralized Deterministic Multi-Agent Reinforcement Learning"
_ICLR.cc/2021/Conference — Reject_

### Official Review · AnonReviewer3 · 2020-10-27
**Detailed theory, however experiments lack coverage**

**Rating:** 5
**Confidence:** 5

**Review:**

Clarity: The paper is well written. The following require some clarifications:
1) $\tilde{a}$ variables are defined by never seem to be used in Algorithm 1, why?

2) In the  off-policy setting (Theorem 1) letting $\mathcal{N}=1$ does not recover the previous result by Silver et al. (January 2014a). This is perhaps due to the fact that eq(1) of this paper has $\bar{R}$ in its objective and eq(14) of Silver et al. (January 2014a) has $Q^{\text{target-policy}}$ instead. It would be great if the authors clarify why the two objectives are different. If my understanding is correct, in the off-policy setting, we are still interested in the maximising the reward with respect to target policy, and the restriction is that the samples are from the behaviour policy.


Quality:
1) Strength: The theoretical results are stated clearly and detailed proofs have been provided. But for the off-policy case, other results seem to be correct.
2) Weakness: The experiments are on a toy domain. The experiments demonstrate that the proposed algorithms indeed converge as expected. However, the experiments do not add any further insights such as a) what happens when we change the communication matrix $C$? b) What happens when the reward structure varies a lot across the agents?  c) what happens when different agents start from different parts of the state space. In short, the experiments have to be carefully thought out to cover the entire range so as to emphasise the multi-agent flavour. As it stands, the current version of the paper spends too little space (only about 4 lines) has been allocated for the experiments.


Novelty:
The deterministic policy gradient result is new. The off-policy case seems to have some issue (see above). While the result in itself is new, it is not clear whether it follows trivially from prior results. It would be great if the authors can comment on the this.


Significance:
The significance of the current version is limited due to,
1) the lack of experiments,
2) the possibility that the new results could be obtained as trivial extension of prior work (Zhang ICML,2018 + Silver et al. (January 2014a)). (I am not saying that this is true, authors can clarify)

---

> ### Author Response · Authors · 2020-11-20
> **Response to Reviewer 3**
>
> We wish to thank Reviewer 3 for the constructive comments and suggestions. The paper has been revised and uploaded. The comments of the reviewer and corresponding changes made are summarized here.
>
> With regards to the clarifications:
>
> (1) As mentioned in the remark following Theorem 5, ã variables are required if compatible features are used for the linear approximation of the Q-function.
>
> (2) We suppose that the reviewer is referring to Theorem 2 (as Theorem 1 is on-policy). This difference from Silver et. al. (January 2014a) comes from the fact that Silver et al. (January 2014a) considers discounted rewards while we consider long-run average reward as the objective to maximize, leading to the formulation of equation (1) used to establish Theorem 2. Note that both setups have been considered and studied in the literature so the new contributions to the long-run average scenario is of interest to the community - deriving deterministic gradient expressions, establishing convergence of actor-critic algorithms and showing that deterministic gradient is a limit case of the stochastic gradient (which had only been shown previously under the discounted rewards objective framework).
>
> With regards to the limited significance:
>
> (1) We have not managed to conduct further experiments yet, but if we manage to do so before the rebuttal deadline, we will post again.
>
> (2) The closest work to ours would be the one of [Zhang ICML,2018], whose convergence proof for the critic step can be adapted straightforwardly step by step to get the convergence of the critic in our deterministic policy case. The difference appears on the actor side as the expression of the deterministic gradient differs from the expression of the stochastic gradient, therefore specific analysis is needed to establish convergence of the policy parameter. Though the overall analysis is a classical but technical two-time-scale algorithm convergence anlaysis, it still requires to be carried out with care to be adapted to the specificities of our update rules. Besides, the proof that the deterministic gradient is a limit case of the stochastic gradient in the context of long-run average reward is not a mere adaptation of [Silver et al. (January 2014a)]'s result obtained for the discounted rewards objective. Looking at the proofs reveal that we are not following the same paths which, e.g.  in our proof, we focus mostly on showing the convergence of stationary distributions and of their gradients (Lemma 2), without considering value functions Q or V, while in  contrary to [Silver et al. (January 2014a)]'s proof establishing convergence of the gradients of Q functions.

---

> > ### Comment · AnonReviewer3 · 2020-11-24
> > **Thanks for the response.**
> >
> > I hereby acknowledge that I have read the author response.

---

### Official Review · AnonReviewer1 · 2020-10-27
**This paper provides a valuable idea and a promising direction in MARL, but the current version has several problems that need to be fixed.**

**Rating:** 4
**Confidence:** 3

**Review:**

This paper extends the results for actor-critic with stochastic policies of [Zhang, ICML 2018] to deterministic policies and offers the proof of convergence under some specific assumptions. The authors consider both the on-policy setting and the off-policy setting and offers some convincing derivation. It provides a valuable idea and a promising direction in MARL, but the current version has several problems that need to be fixed. Specifically, some parts of equations, algorithms, and expressions are ambiguous and unintelligible. Besides, problems with the format in the formula and citations also exist, which degrade the paper’s quality and clarity.

Pros:
1.	The motivation that considers both off-policy and on-policy is interesting and attractable. This work might provide a promising way to address the problem of inefficiency of exploration in MARL.
2.	The convergence of MARL is challenging, and this paper gives a valuable attempt.

Cons:
1.	The experiment is insufficient and doubtful, although this paper is convincing in theory, the experiments are not convincing at all. The convergence curves are quite nice, but the authors should compare the results with some other state-of-art algorithms to increase its credibility.
2.	The differences with [Zhang, ICML 2018] are ambiguous. Some definitions and formulas and even algorithms are almost the same as [Zhang, ICML 2018].
3.	There are lots of format problems. For example, “Kaiqing Zhang, Zhuoran Yang, Han Liu, Tong Zhang, and Tamer Basar. Fully decentralized multiagent reinforcement learning with networked agents. 80:5872–5881, 10–15 Jul 2018.” In it, I cannot find where this paper is published. Some symbols are used without explanation, e.g., t in Algorithms 1 and 2.
4.	The organization of this paper needs to be polished. In the background section, the authors firstly introduce the optimization problem of maximizing the average reward with the objective $J(\pi_\theta)$ on page 2. However, they use $J(\theta)$ when defining the Q_\theta function instead of $J(\pi_\theta)$ just 3 lines below. This simplification is not noted until Section 3. So the authors should re-organize the paper to make the content more readable.
5.	This paper raises some assumptions when discussing, but neither does it offer a list of these assumptions, nor does it analyze the rationality and influence of them.

Questions:
1.	What is the difference between Figure 1 and Figure 2?
2.	What is the detail of the experiment?
3.	How well do the results compared with [Zhang, ICML 2018]?
4.	In Algorithm 1, why can you draw an action firstly?

---

> ### Author Response · Authors · 2020-11-20
> **Response to Reviewer 1**
>
> We wish to thank Reviewer 1 for the constructive comments and suggestions. The paper has been revised and uploaded. The comments of the reviewer and corresponding changes made are summarized here.
>
> With regards to the cons,
>
> (1) We have not managed to conduct further experiments yet, but if we manage to do so before the rebuttal deadline, we will post again.
>
> (2) There is significant differences from [Zhang, ICML 2018] as the latter handles discrete action space while our paper handles continuous action space. Although their convergence proof for the critic step can be adapted step by step to get the convergence of the critic in our setup, there is a significant difference on the actor side as the expression of the deterministic gradient differs from the expression of the stochastic gradient, therefore more specific analysis is needed to establish convergence of the policy parameter. Though the overall analysis approach follows classical two-time-scale algorithm convergence analysis, it is significantly technical, and therefore still needs to be carried out with care to be adapted to the specifities of our updates. In addition, we also prove that the deterministic gradient is a limit case of the stochastic gradient in the context of long-run average reward, which is not a straightforward adaptation of [Silver et al. (January 2014a)]'s result for the discounted reward setting.
>
> (3) We apologise for the typo. This is the reference to the ICML paper and we has been fixed in the revision. We have also properly introduced and updated the step variable t in the algorithms.
>
> (4) We apologise for the confusion. These are the same quantities.
>
> (5) The full list of assumptions were originally provided in the Appendix. In the revision, these have been moved from the Appendix to the main body. These are technical assumptions on the parameters of the algorithm and do not impose conditions on the problem setting. The assumptions on the problem setting are described in Section 2, notably the regularity assumptions on the Markov chain, and these are standard assumptions in the literature.
>
> With regards to the questions,
>
> (1) Yes Figure 2 was a reproduction of Figure 1 for completeness of the Appendix.
>
> (2) The details of the experiment are provided in the Appendix.
>
> (3) The results are not comparable with [Zhang, ICML 2018] as the latter is for discrete action space, while our work is for continuous action space.
>
> (4) Unfortunately, we do not completely understand the concern on drawing an action. At the initial step, the state and policy parameters have been initialised, so an action can be obtained. In each iteration, an action is drawn as part of the typical exploration of the RL algorithm.

---

### Official Review · AnonReviewer2 · 2020-11-03
**comprehensive but somewhat incremental**

**Rating:** 6
**Confidence:** 3

**Review:**

This paper offers a comprehensive theoretical treatment of deterministic policy gradients in a multi-agent setting, working out several key results:
* existence and explicit formulas for the multi-agent deterministic policy gradient in off and on-policy settings;
* convergence of stochastic policy gradients to deterministic ones as policy variance converges to zero;
* convergence of multi-agent deterministic actor-critic algorithms.

This paper is very well-written and easy to follow, although technically there is not much to expose here, as most of the paper consists of statements of results. I did not check all the math thoroughly, but there does not seem to be any cause for suspicion as far as I can tell. Related work is sufficiently well presented.

I think that the topic of deterministic policy gradient in a multi-agent is sufficiently interesting and having the foundational results worked out in a single reference is a valuable contribution. The paper is incremental in a way that the deterministic results seem to be extensions of known stochastic results along the lines of well-understood techniques. If any unexpected challenges had to be addressed in order to derive these results (e.g. developing new techniques), this has to be made clearer in the paper.

Overall I think this is a solid paper.

---

> ### Author Response · Authors · 2020-11-20
> **Response to Reviewer 2**
>
> We wish to thank Reviewer 2 for the constructive comments and suggestions.
>
> We would like to address the comment that the work might be incremental. The closest work to ours would be the one of [Zhang ICML,2018], whose convergence proof for the critic step can be adapted step by step to get the convergence of the critic in our setup. However, there is a significant difference on the actor side as the expression of the deterministic gradient differs from the expression of the stochastic gradient, therefore more specific analysis is needed to establish convergence of the policy parameter. Though the overall analysis approach follows classical two-time-scale algorithm convergence analysis, it is significantly technical, and therefore still needs to be carried out with care to be adapted to the specifities of our updates. In addition, we also prove that the deterministic gradient is a limit case of the stochastic gradient in the context of long-run average reward, which is not a straightforward adaptation of [Silver et al. (January 2014a)]'s result for the discounted reward setting.

---

### Official Review · AnonReviewer4 · 2020-11-05
**This paper proposes two algorithms for learning deterministic policy in multi-agent RL problem. The authors extend the multi-agent policy gradient for stochastic policies to deterministic policies. Decentralized algorithms are proposed to learn the Q/averaged reward in on/off-policy setting.**

**Rating:** 5
**Confidence:** 4

**Review:**

This paper proposes a solution to learning deterministic policy in the multi-agent RL setting, where local rewards are private to local agents.  Both on-policy learning and off-policy learning are considered in the paper.
To do so, the author first extends the policy gradient theorem in MARL for stochastic policy to deterministic policy.
Specifically, in on-policy learning, only global Q-function and local action are needed to compute local policy gradient; in off-policy setting, only averaged reward function and local action are needed.
The author then propose a decentralized algorithm for learning the global Q-function/averaged reward function. Two-timescale technique is adopted to show the convergence of the proposed algorithms.

Strengths:
(1) the extension of policy gradient to deterministic policy in MARL (locally observable reward) is interesting and important.
(2) the decentralized algorithm itself, although similar to prior work in Zhang et al (2018), provide needed guarantee to the algorithm for convergence.
(3) the paper is well written and easy to follow.

Cons:
The proposed decentralized algorithm closely mimics the one in the prior work, in addition to its analysis framework.  I hope the authors can provide more details on how the analysis of the algorithm differentiates itself from previous work in different aspects. I am happy to increase my rating if the author can address this issue in their response.

---

> ### Author Response · Authors · 2020-11-20
> **Response to Reviewer 4**
>
> We wish to thank Reviewer 4 for the constructive comments and suggestions.
>
> The closest work to ours would be the one of [Zhang ICML,2018], whose convergence proof for the critic step can be adapted step by step to get the convergence of the critic in our setup. However, there is a significant difference on the actor side as the expression of the deterministic gradient differs from the expression of the stochastic gradient, therefore more specific analysis is needed to establish convergence of the policy parameter. Though the overall analysis approach follows classical two-time-scale algorithm convergence analysis, it is significantly technical, and therefore still needs to be carried out with care to be adapted to the specifities of our updates. In addition, we also prove that the deterministic gradient is a limit case of the stochastic gradient in the context of long-run average reward, which is not a straightforward adaptation of [Silver et al. (January 2014a)]'s result for the discounted reward setting.

---

### Official Review · AnonReviewer5 · 2020-11-06
**Meaningful results but not strong enough**

**Rating:** 5
**Confidence:** 4

**Review:**

This paper establishes the asymptotic convergence of on- and  off- policy DPG in the multi-agent setting under some assumptions. Overall the paper is well written and easy to follow. Given the practical usefulness of DPG, I think the result in this paper is somehow meaningful to the community.

However, the paper has the following issues:
(1) Since this paper is a theoretical paper, the assumptions are very important to evaluate the signficance of the main theorems. Thus, the assumptions should be moved to the main body of the paper.

(2) The projection is a practical issue that has been criticized by many researchers in the RL community. Although it is necessarily to make this assumption when handling two time-scale algorithm with time-varying MDP, it can be avoid by adopting other updating structures such as nested-loop, mini-batch two time-scale, etc. Or can the author provide an upper for such a projection radius so that the algorithm can at least be implemented in practice?

(3) Although this paper is a theoretical paper, only considering multi-agent bandit in the experiement is not sufficient to verify the theoretical results in this paper. I suggest the author to consider some RL settings to make the experiment results stronger.

(4) The current muti-agent setting in this paper is not that practical, as too many variables are requied to be globally observable.

(5) The update at time step t required the knowledge in step (t+1), which could introduce some pratice issues as this paper focus on online update. Can the author propose some potential idea to fix this issue?

=== After rebuttal ===

Since the technical contribution of this paper is not significant enough, I will keep my score as weakly reject.

---

> ### Author Response · Authors · 2020-11-20
> **Response to Reviewer 5**
>
> We wish to thank Reviewer 5 for the constructive comments and suggestions. The paper has been revised and uploaded. The comments of the reviewer and corresponding changes made are summarized here.
>
> (1) The assumptions have been moved to the main body.
>
> (2) The projection step has been often used to show convergence in several papers. Some even directly assume boundedness of the update leading to easier convergence analysis - notably in [Bhatnagar et al., Automatica (2009)] where they mention that in practice they do not do the projection but observe empirically that the updates remain bounded and converge.
>
> (3) We have not managed to conduct further experiments yet, but if we manage to do so before the rebuttal deadline, we will post again.
>
> (4) While the exact algorithm might indeed require global observability, nevertheless, the algorithm provides intuition for designing algorithms with partial observability, for example in settings where the influence between agents decays with some intuitive measure of distance in the communication graph.
>
> (5) Unfortunately, we do not understand this comment that the update at step t requires knowledge in step t+1. In each time step, the actions are taken, and then the update is performed based on the actions that are taken. Moreover, in practice, it is also common to accumulate RL updates into batches, or use target networks that are updated more slowly. While we do not provide convergence guarantees under such practical heuristic modifications, these tricks are typically observed to lead to performance and stability improvements.

---

### Decision · Program_Chairs · 2021-01-07
**Final Decision**

**Decision:**

Reject

**Comment:**

The paper offers a direction for mult-agent RL that builds on results for actor-critic methods [Zhang, ICML 2018], extending that work to address deterministic policies.  The authors establish convergence under a number of assumptions.   Both on-policy setting and off-policy settings are treated.  The reviewers point out several concerns and the consensus seems to be that, while the direction looks promising, the paper deserves further work.